# MYADM binds human parechovirus 1 and is essential for viral entry

Wenjie Qiao [1], Christopher M. Richards[1], Youlim Kim [1], James R. Zengel[1], Siyuan Ding[2], Harry B. Greenberg[1,3,4] & Jan E. Carette [1] ✉

Human parechoviruses (PeV-A) are increasingly being recognized as a cause of infection in neonates and young infants, leading to a spectrum of clinical manifestations ranging from mild gastrointestinal and respiratory illnesses to severe sepsis and meningitis. However, the host factors required for parechovirus entry and infection remain poorly characterized. Here, using genome-wide CRISPR/Cas9 loss-of-function screens, we identify myeloid-associated differentiation marker (MYADM) as a host factor essential for the entry of several human parechovirus genotypes including PeV-A1, PeV-A2 and PeV-A3. Genetic knockout of MYADM confers resistance to PeV-A infection in cell lines and in human gastrointestinal epithelial organoids. Using immunoprecipitation, we show that MYADM binds to PeV-A1 particles via its fourth extracellular loop, and we identify critical amino acid residues within the loop that mediate binding and infection. The demonstrated interaction between MYADM and PeV-A1, and its importance specifically for viral entry, suggest that MYADM is a virus receptor. Knockout of MYADM does not reduce PeV-A1 attachment to cells pointing to a role at the post-attachment stage. Our study suggests that MYADM is a multi-genotype receptor for human parechoviruses with potential as an antiviral target to combat disease associated with emerging parechoviruses.

Human parechoviruses (PeV-A), belong to the *Picornaviridae*, a group of RNA viruses of positive strand polarity, which includes poliovirus and other enteroviruses. To date, 19 different PeV-A genotypes have been identified, with PeV-A1 and PeV-A3 being more prevalent[1]. Parechoviruses are a frequent cause of infection in young children and generally cause mild gastrointestinal or respiratory symptoms[2]. However, PeV-A3 is increasingly recognized as causing severe disease including neonatal sepsis, childhood meningitis, and encephalitis[3–8]. Despite the clinical relevance of parechoviruses as emerging pathogens, cellular factors required for viral entry and replication are largely uncharacterized.

Picornaviruses use host-encoded receptors to mediate attachment to the cell and facilitate conditions favorable to cross the cellular membrane and deliver their genome to the cytosol[9]. Cellular receptors are key determinants of tissue tropism and play direct roles in viral pathology[10]. PeV-A1 interacts with its cellular receptor integrin through the arginine-glycine-aspartic acid (RGD) peptide motif in its capsid, which is also present in several other PeV-A genotypes[11,12]. Whether integrin acts in concert with other receptors is unknown. Moreover, the identity of essential receptors for genotypes such as PeV-A3 that lack the RGD motif is enigmatic.

Here, we use genome-scale CRISPR/Cas9 screening in human cells and identify myeloid-associated differentiation marker (MYADM) as an essential host factor for PeV-A1, PeV-A2 and PeV-A3 infection. MYADM is a broadly expressed plasma membrane protein that partitions in detergent-resistant membranes suggesting its association with

[1]Department of Microbiology and Immunology, Stanford University School of Medicine, Stanford, CA, USA. [2]Department of Molecular Microbiology, Washington University School of Medicine, St. Louis, MO, USA. [3]Division of Gastroenterology and Hepatology, Department of Medicine, Stanford University School of Medicine, Stanford, CA, USA. [4]Department of Veterans Affairs, VA Palo Alto Health Care System, Palo Alto, CA, USA. ✉e-mail: carette@stanford.edu

membrane rafts[13,14]. We show that MYADM is required for viral entry in cell lines and confirm that MYADM mediates infection in human gastrointestinal epithelial organoids representing a primary site of PeV-A infection[7,8]. Using mutagenesis based on the divergent ability of human parechovirus to engage with MYADM from several mammalian species, we identify specific amino acid residues within the fourth extracellular loop of MYADM as critical for virus infection. While MYADM does not determine initial cell binding for PeV-A1, we show that MYADM binds to PeV-A1 particles pointing to a role as essential post-attachment receptor.

## Results

### MYADM is a host factor required for integrin-dependent and -independent parechoviruses

To identify cellular factors essential for parechovirus infection, we performed CRISPR/Cas9 genetic screens in human cells (Fig. 1a). Because parechoviruses are known to infect the intestinal tract, we selected HT-29 cells derived from primary colorectal adenocarcinoma as the cell line for the pooled CRISPR/Cas9 screens. To favor identification of cognate protein receptors[15], we modified HT-29 cells to prevent cell surface expression of heparan sulfate and sialic acid by knockout of critical enzymes in the biosynthetic pathway of these glycans. These cells contain mutations in EXTL3 and SLC35A1 and are referred to as HT-29 double knockout cells (HT29-DKO, Supplementary Fig. 1). After introduction of a genome-scale lentiviral library[16], we infected the cells with PeV-A1 or PeV-A2. The cell population resistant to cytolytic PeV replication was harvested, and the guide RNA

sequences (gRNAs) were amplified from genomic DNA. The identity and abundance of the gRNAs were determined through sequencing and compared to the uninfected parental cell library. *MYADM* and integrin subunit beta 6 (*ITGB6*) were identified as genes with the most enriched gRNAs in both PeV-A1 and PeV-A2 genome-scale screens, and *ITGAV* was identified within the top 10 highest scoring genes (Fig. 1a, Supplementary Data 1). Thus, the genome-scale screens confirmed integrins, specifically alpha(V)beta(6) integrin, as receptors for PeV-A1 and PeV-A2[12,17], and nominated the plasma-membrane protein MYADM as a candidate essential receptor.

To validate the top candidate, we generated isogenic cell lines with a MYADM knockout mutation (ΔMYADM, Supplementary Figs. 2, 3). We used HT29-DKO cells and another intestinal cell line susceptible to parechovirus infection, HuTu80[18]. To measure PeV-A1 infection over a time course, we used PeV-A1 engineered to express nanoluciferase (PeV-A1-nLuc, Supplementary Fig. 4). Infection of PeV-A1-nLuc was prevented in ΔMYADM cells, and trans-complementation of MYADM restored infectivity to levels comparable to wild type cells (Fig. 1b). The same inability to support infection was observed in several ΔMYADM cell lines (HuTu80, 293FT and A549 cells, Supplementary Figs. 2, 3) using unmodified PeV-A1 and measuring RNA replication using an RT-qPCR assay (Fig. 1c, d). Notably, ΔMYADM cells were also refractory to infection by PeV-A3, indicating that MYADM is essential for this integrin-independent strain (Fig. 1c, d). The dependence of PeV-A1 infection, but not PeV-A3 infection, on integrins was confirmed in ΔITGB6 A549 cells (Fig. 1d). Trans-complementation in ΔMYADM HuTu80 and 293FT cells using stable over-expression resulted in

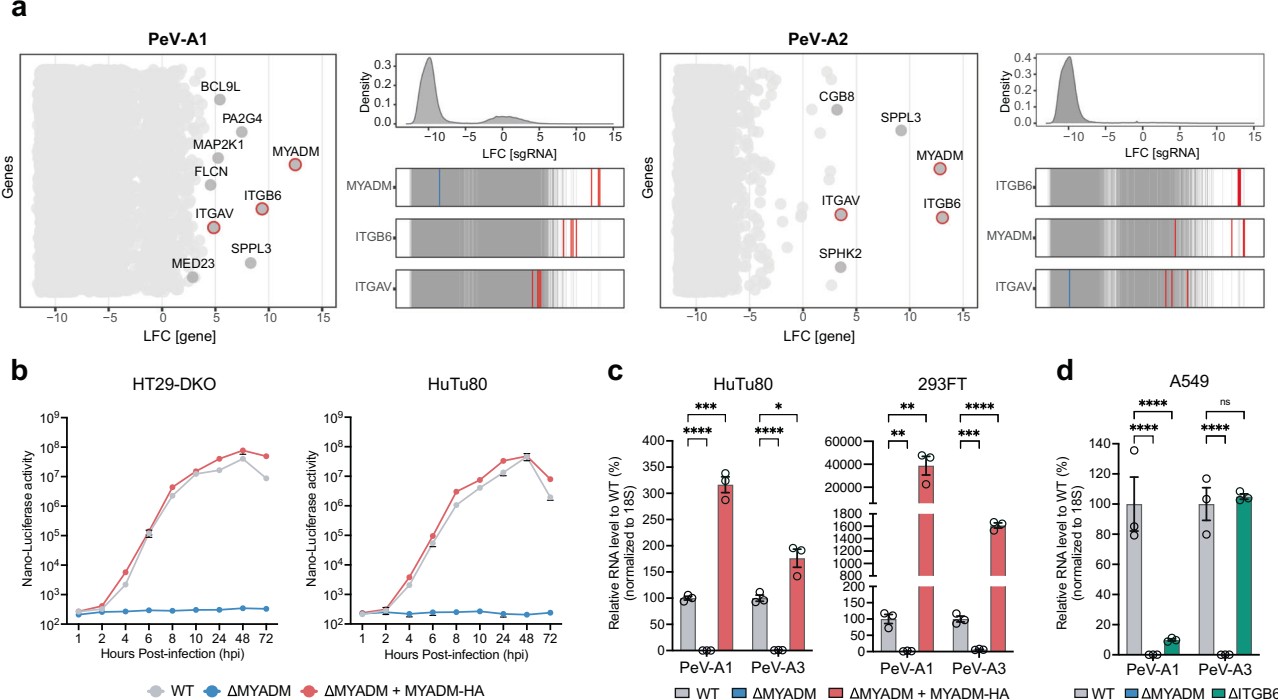

**Fig. 1 | MYADM is an essential host factor for human parechoviruses. a** CRISPR screens for human parechovirus A1 (PeV-A1) and A2 (PeV-A2) in heparan-sulfate/sialic acid deficient HT-29 cells (HT29-DKO, ΔEXTL3 and ΔSLC35A1). Left: Bubble plot illustrating enriched genes based on log2(fold change) (LFC) in the PeV-A1 or PeV-A2-selected population, with genes having FDR < 0.01 depicted in dark gray. Right: Distribution of LFC of sgRNAs from the entire library (top), and sgRNAs enrichment targeting indicated genes overlaid on gray lines representing the distribution of all sgRNAs (bottom). **b** Time course of nLuc-expressing PeV-A1 infection in HT29-DKO and HuTu80 wild type (WT), MYADM knockout (ΔMYADM), and HA-tagged MYADM-complemented MYADM knockout (ΔMYADM + MYADM-HA) cells at MOI 0.1. Data are presented as mean values +/− SD, n = 3 biologically

independent samples. **c** RT-qPCR quantification of PeV-A1 and PeV-A3 RNA in infected HuTu80 and 293FT WT, ΔMYADM, and ΔMYADM + MYADM-HA cells at MOI 0.1 at 8 hpi or 48 hpi, respectively. Data are presented as mean values +/− SEM, n = 3 biologically independent samples, multiple two-sided unpaired t tests with the Holm–Šídák's method for multiple comparison, *P < 0.05; **P < 0.01; ***P < 0.001; ****P < 0.0001. **d** RT-qPCR quantification of PeV-A1 and PeV-A3 RNA in infected A549 WT, ΔMYADM, and ITGB6 knockout (ΔITGB6) cells at MOI 0.1 at 48 hpi. Data are presented as mean values +/− SEM, n = 3 biologically independent samples, two-way ANOVA with Šídák's multiple comparison test, ****P < 0.0001; ns not significant. Source data are provided as a Source Data file.

markedly enhanced susceptibility compared to wild type cells (Fig. 1c). These results were further confirmed with plaque assays (Supplementary Fig. 5). To address the specificity of MYADM deletion on pan-picornavirus replication, we tested EV-A71 and CV-B3 reporter viruses in wild type and ΔMYADM 293FT cells. No significant effect on virus replication was observed in ΔMYADM cells (Supplementary Fig. 6). Together, these results show that MYADM is essential for infection by multiple PeV-A genotypes and that MYADM expression can be rate limiting for infection.

## MYADM is essential for infection of primary intestinal organoids
Like enteroviruses, human parechoviruses are frequently detected in stool samples pointing to fecal-oral transmission as a major route of infection[8]. Previously, primary intestinal organoids have been used as biologically relevant in vitro models representing the intestinal epithelium relevant for these enteric pathogens[19–21]. We used adult stem cell-derived 3D intestinal organoids where the apical surface faces outward to the medium[22]. We generated two independent isogenic clonal stem cell lines with a genetic knockout mutation in MYADM using CRISPR/Cas9 (Supplementary Fig. 7). Wild type organoids were susceptible to PeV-A1 infection as measured by immunostaining for double-stranded RNA, which marks cells with replicating RNA (Fig. 2a). In stark contrast, ΔMYADM organoids were refractory to infection. Infection with the unrelated virus herpes simplex virus (HSV) was not affected by MYADM knockout (Fig. 2a). In wild type organoids PeV-A1 viral RNA levels and infectious particles increased strongly during a time-course of infection, while no such increase was observed in ΔMYADM organoids (Fig. 2b). No replication defect was observed for HSV-GFP (Fig. 2c). Compared to PeV-A1, PeV-A2 exhibited slower kinetics in wild type colonoids, with no detectable infection observed in ΔMYADM organoids (Supplementary Fig. 8a). Additionally, a notable increase in infection was observed in wild type colonoids exposing the basolateral side with both PeV-A1 and A2 compared to the apical-out colonoids, while ΔMYADM organoids still showed no increase in infectious particles (Supplementary Fig. 8b). Thus, MYADM is a critical factor for PeV-A1 and A2 infection in multiple human cell lines as well as in primary epithelial organoids derived from human intestinal tissue.

## MYADM determines host tropism via its extracellular loop
Receptor usage is often a critical determinant in the ability of viruses to infect different host species, which can complicate the development of rodent models of pathogenesis by human viruses including picornaviruses and coronaviruses[10,23]. To investigate whether MYADM might act similarly, we expressed human MYADM in the wild type BHK21 cell line, derived from hamster, and in nontransformed wild type mouse embryonic fibroblasts (MEFs). We tested viral infection by using PeV-A1 engineered to express the mNeonGreen fluorescent protein (PeV-A1-GFP, Supplementary Fig. 4), and live cell imaging to quantify the infection over time. While both rodent wild type cells were refractory to PeV-A1-GFP infection, expression of human MYADM rendered both cell lines susceptible to infection (Fig. 3a, Supplementary Fig. 9a). In contrast, overexpression of human ITGB6 did not confer susceptibility (Fig. 3a, Supplementary Fig. 9a). Similar results were obtained using PeV-A1-nLuc infection (Supplementary Fig. 9b). To assess the ability of MYADM proteins from distinct mammalian species to act in human cells as essential host factors, we complemented ΔMYADM cells with MYADM via lentiviral transduction (Fig. 3b, c, d). Consistent with the inability of MEF cells to support infection, mouse MYADM did not rescue PeV-A1 infection (Fig. 3c). MYADM from species whose amino acid sequence were more closely related to human (i.e. macaque, cat and dog) acted to functionally restore infection, while more distant species (i.e. mouse, bovine, horse, ferret and goat) failed to do so (Fig. 3b, c). MYADM is a plasma membrane protein that has eight predicted transmembrane helices with cytoplasmic N- and C-terminal regions[13,24]. Because its last extracellular loop is divergent between

species (Fig. 3e, Supplementary Fig. 10), we generated chimeric constructs between human and hamster MYADM exchanging this loop region (Fig. 3f). Strikingly, by introduction of the human extracellular loop region in hamster MYADM, it gained the ability to support PeV-A1 infection (Fig. 3f). Conversely, human MYADM with the hamster extracellular loop lost its ability to mediate infection (Fig. 3f). To further narrow down the essential amino acids within MYADM's loop that determine the host tropism, we focused on the amino acid differences in the species that lost the ability to act as PeV-A1 host factor (Fig. 3e). Horse MYADM has the least residues that are divergent from human MYADM. From the six amino acids that differed, mutation analysis pinpointed three amino acids of critical importance (Supplementary Fig. 11). Changing the three amino acids in human MYADM to the horse sequence negated the ability to support infection, while changing these residues in horse MYADM to the human sequence allowed virus replication at levels comparable to human MYADM as determined using PeV-A1-nLuc (Fig. 3g, Supplementary Fig. 11). Mutagenesis of single amino acids further pinpointed this phenotype to one residue (V in human and H in horse), although the effect was less pronounced compared to mutagenesis of all three amino acids (Fig. 3g). Similar results were obtained using PeV-A1-GFP as shown by live cell microscopy (Supplementary Fig. 12). Together, these experiments suggest that MYADM can determine species tropism of human parechoviruses and identify the last extracellular loop as a critical determinant of MYADM's function as a host factor.

## MYADM is not essential for viral RNA replication
To investigate at which step in the viral life cycle MYADM acts, we made use of a replicon system. For this, we modified the fully infectious clone PeV-A1-nLuc by removing part of the sequence encoding the capsid proteins (PeV-A1-nLuc replicon) (Fig. 4a). Transfection of in vitro transcribed replicon RNA bypasses viral entry but allows viral translation and RNA replication to occur (Fig. 4a). While the ability of mouse cells (MEFs) to support PeV-A1-nLuc infection was strictly dependent on the expression of human MYADM when infected with PeV-A1-nLuc particles, no such dependence was found upon transfection with PeV-A1-nLuc replicon RNA (Fig. 4b). Likewise, knockout of MYADM in human cells (293FT) did not decrease replication of the PeV-A1-nLuc replicon nor did MYADM overexpression enhance it, which was in stark contrast to infection using PeV-A1-nLuc virus (Fig. 4b). From this we conclude that MYADM is not affecting viral RNA replication, which indicates an essential role in viral entry.

## MYADM interacts with parechovirus particles
Picornavirus receptors mediate attachment of the virus to the cell and trigger the release of viral RNA into the cytosol. The latter step occurs post-attachment, and is mediated by interactions of the receptor with the viral capsid[9,25]. For some picornaviruses, cell attachment and RNA release are mediated by one receptor, while in other cases these separate steps are mediated by distinct receptors[10]. To test the potential role of integrin and MYADM on PeV-A1 cell attachment, we determined the initial binding of virus to cells at 4 °C and the subsequent internalization after 1 h incubation at 37 °C. Knockout of the integrin subunit ITGB6 reduced binding, internalization and infection, which could be restored upon stable expression of ITGB6 (Fig. 5a, b). In contrast, while MYADM knockout drastically reduced infection, it did not prevent binding and internalization of PeV-A1 in A549 cells (Fig. 5a, b). Similarly, stable overexpression of human ITGB6 but not MYADM in MEF cells increased cell binding and internalization (Supplementary Fig. 13). The inability of MYADM knockout to affect PeV-A1 cell attachment and internalization was also observed in HT29-DKO cells using fluorescent in situ hybridization (Supplementary Fig. 14).

To test whether MYADM binds to PeV-A1 particles during cell entry, we infected cells expressing human MYADM tagged with the HA epitope, which allows for immunoprecipitation. As a control we

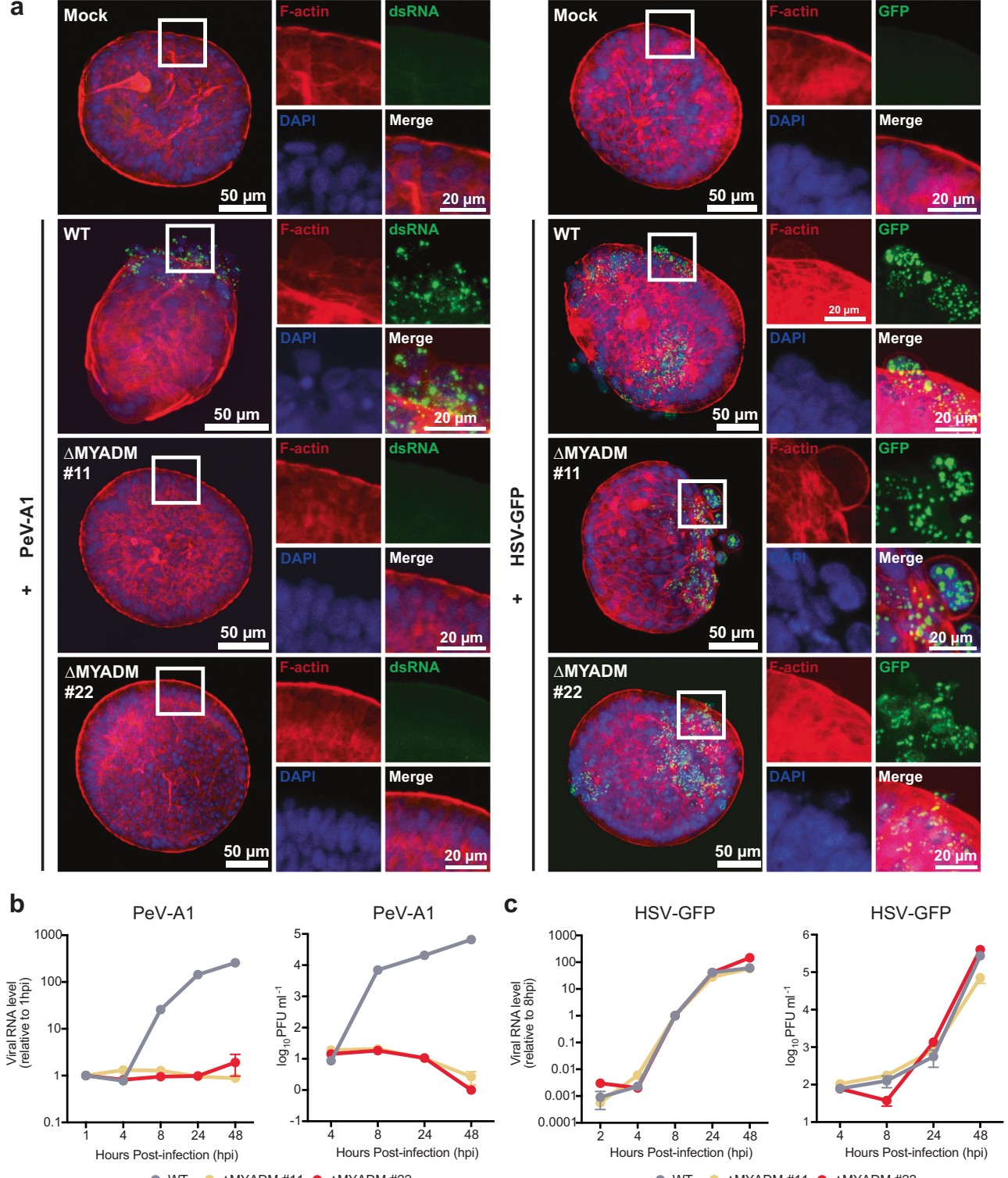

**Fig. 2 | MYADM is required for infection of primary intestinal organoids. a** Wild type (WT) and two MYADM knockout human colonoid lines (ΔMYADM #11 and ΔMYADM #22) were infected with PeV-A1 or K26GFP (HSV-GFP) and stained for double-stranded RNA or GFP, respectively. Uninfected colonoids (Mock) served as negative controls. Right panels illustrating increased magnification of square boxes in the left panel. **b** Quantification of PeV-A1 infection in colonoids using RT-qPCR and plaque assay over time. Data are presented as mean values +/− SEM, n = 3 or 4 biologically independent samples. **c** Quantification of K26GFP (HSV-GFP) infection in colonoids using RT-qPCR and plaque assay over time. Data are presented as mean values +/− SEM, n = 3 biologically independent samples. Source data are provided as a Source Data file.

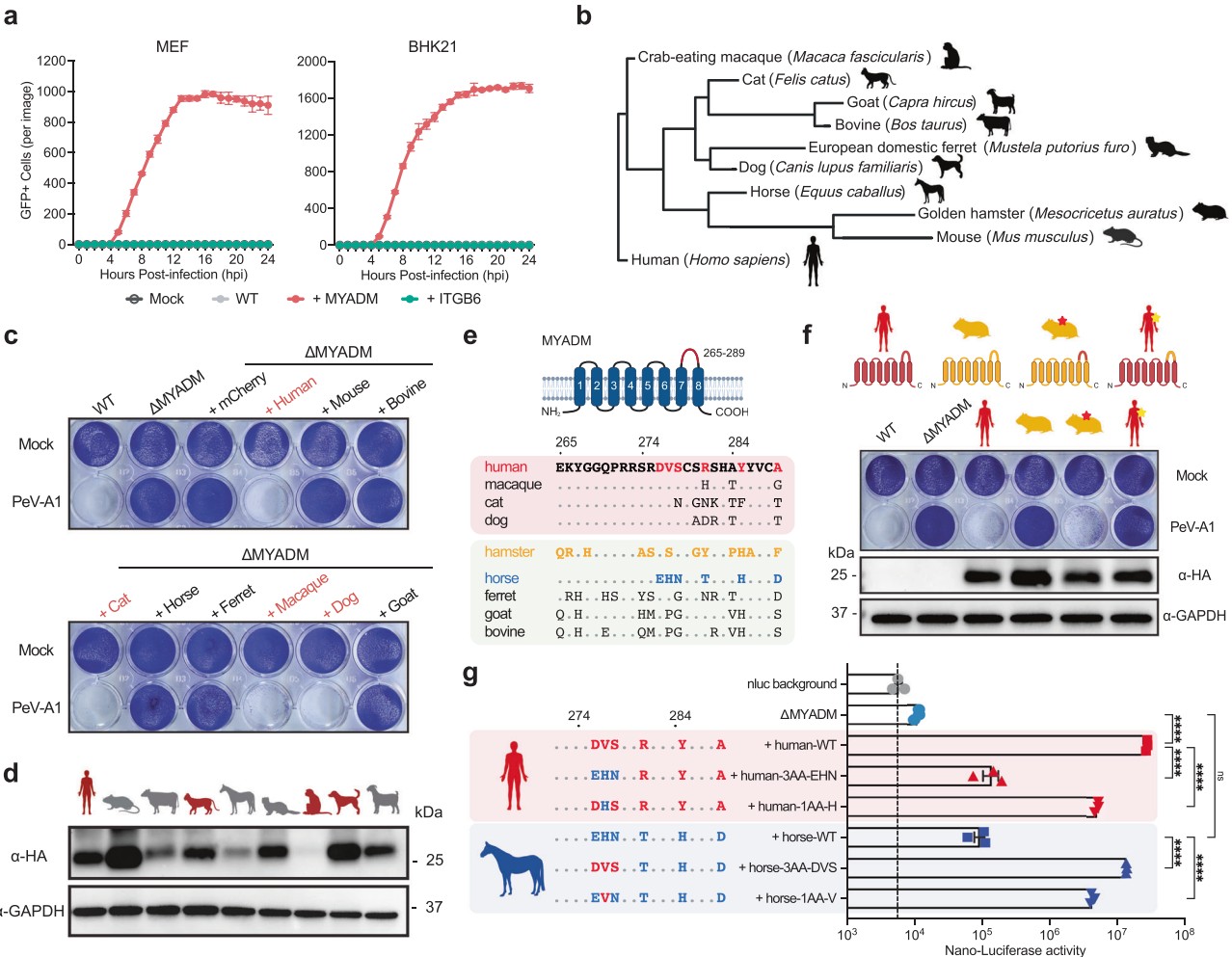

**Fig. 3 | MYADM determines host tropism via its extracellular loop. a** Time course of GFP-expressing PeV-A1 infection in wild type (WT), human MYADM-expressing (+ MYADM), and human ITGB6-expressing (+ITGB6) MEF/BHK21 cells at MOI 1. Data are presented as mean values +/−SEM, $n$ = 3 biologically independent samples. Uninfected cells (Mock) served as negative controls. **b** Cladogram of full-length MYADM proteins from distinct mammalian species. **c** Crystal violet staining of HT29-DKO WT, MYADM knockout (ΔMYADM), and ΔMYADM cells expressing HA-tagged MYADM from various mammalian species, infected with PeV-A1 at MOI 0.5. **d** Western blot analysis showing the expression of HA-tagged MYADM from various mammalian species in HT29-DKO ΔMYADM cells. GAPDH was blotted as a loading control. Data from one experiment representative of two independent experiments ($N$ = 2) are shown. **e** Top: Schematic representation of the human MYADM protein structure. Bottom: Alignment of the divergent region of the fourth extracellular loop of MYADM proteins, highlighting the distinct amino acids from

human MYADM. **f** Top: Schematic of the WT and extracellular domain swap structures of human and hamster MYADM, i.e., amino acids 265-289. Bottom: Crystal violet staining of HT29-DKO WT, ΔMYADM, and ΔMYADM cells expressing HA-tagged human/hamster MYADM proteins, infected with PeV-A1 at MOI 0.5. Protein expression was quantified by western blot analysis. Data from one experiment representative of two independent experiments ($N$ = 2) are shown. **g** Luciferase expression at 24 hpi following PeV-A1-nLuc infection (MOI 0.1) in HT29-DKO ΔMYADM cells, ΔMYADM cells expressing human MYADM or horse MYADM WT, and corresponding mutants depicted on the left panel. PeV-A1-nLuc only without cells was used as nLuc background control. Data are presented as mean values +/− SEM, $n$ = 3 biologically independent samples, one-way ANOVA with Šídák's multiple comparison test, ****$P$ < 0.0001; ns not significant. Source data are provided as a Source Data file.

expressed the 3AA-EHN mutant, which contains mutations in the extracellular loop that negate the function of MYADM to act as PeV-A1 host factor (Fig. 3g). PeV-A1 particles were quantified using RT-qPCR from cell lysates before ("input") or after immunoprecipitation ("IP"). No differences were observed in the input consistent with the ability of PeV-A1 virus to attach and internalize independent of functional MYADM (Fig. 5a, Supplementary Fig. 15). However, after immunoprecipitation, we observed a drastic enrichment of PeV-A1 RNA specifically for wild type MYADM in cells allowed to internalize virus at 37 °C for 15 and 30 min (Fig. 5c, Supplementary Fig. 15), while such enrichment was not detected with EV-A71 virus (Supplementary Fig. 16). This co-immunoprecipitation suggests that PeV-A1 particles can bind to MYADM early during viral entry especially at time points corresponding to endocytosis. The gradual acidification in the endosomal

pathway, in conjunction with receptor-virus interactions, is exploited by several picornaviruses to trigger the release of viral RNA into the cytosol[25,26]. To test binding between MYADM and PeV-A1 particles more directly, we engineered a Flag-tagged PeV-A1 by fusing the epitope tag to the structural protein VP1 (PeV-A1-Flag, Supplementary Fig. 17). We immunoprecipitated MYADM from uninfected cells and incubated it with concentrated PeV-A1-Flag particles (Fig. 5d). Unbound particles were removed through washing steps. We observed binding that was specific for wild type MYADM under slightly acidic conditions but not at slightly alkaline pH (Fig. 5e, f). No binding was detected between MYADM and EV-A71 (Supplementary Fig. 18). The specific binding of MYADM to PeV-A1 particles during entry and in an in vitro assay suggests that MYADM acts as an essential receptor for human parechovirus (Fig. 5g).

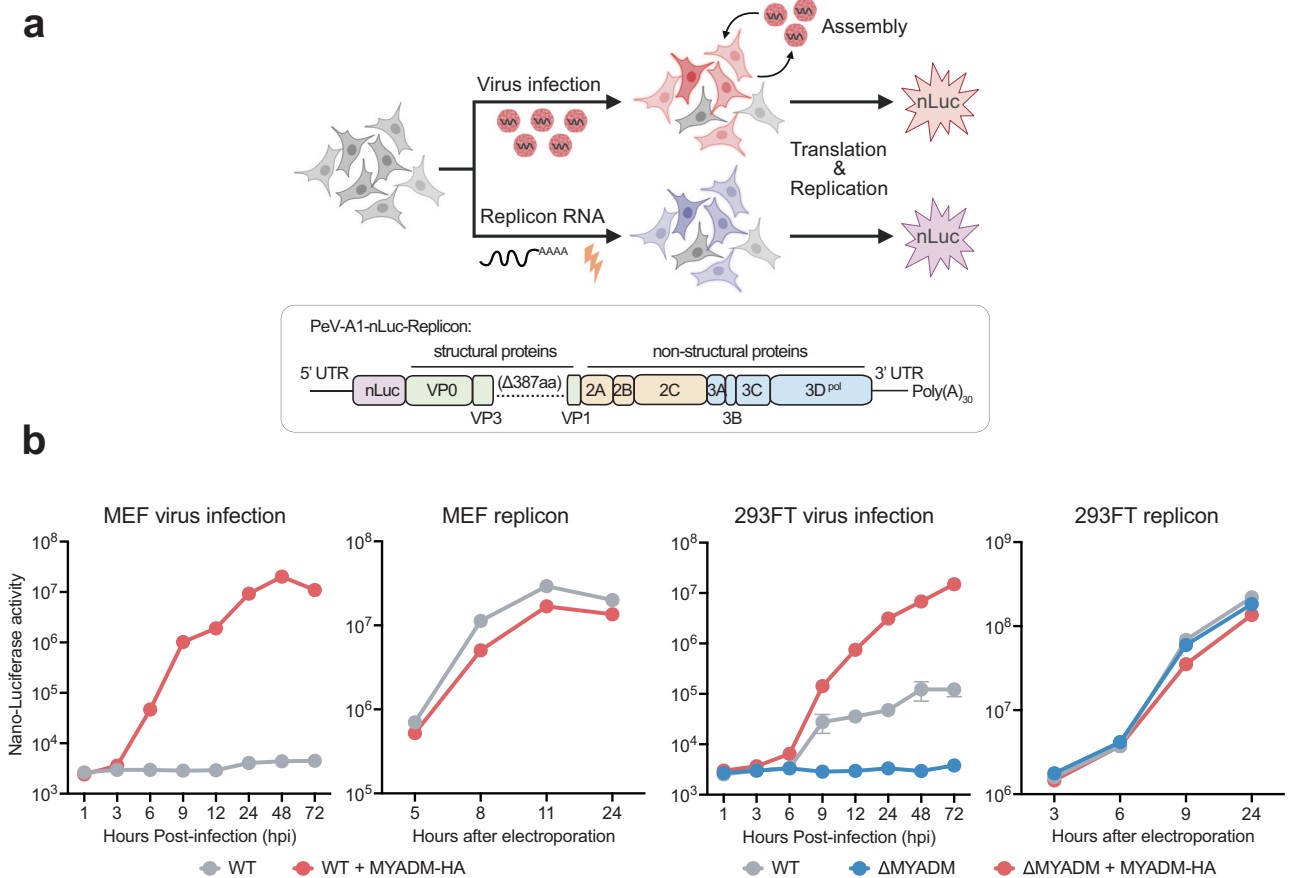

**Fig. 4 | MYADM is not essential for viral RNA replication. a** Top: Schematic illustrating the comparison between virus infection and replicon assay, where replicon RNA was delivered by electroporation. Bottom: Schematic of the PeV-A1-nLuc-Replicon construct. **b** Time course of PeV-A1-nLuc virus infection and PeV-A1-nLuc-Replicon assay in MEF (WT: wild type; WT + MYADM-HA: MEF expressing human MYADM-HA) and 293FT (WT: wild type, ΔMYADM: MYADM knockout, ΔMYADM + MYADM-HA: human MYADM-HA-complemented MYADM knockout) cell lines. Data are presented as mean values +/− SEM, n = 3 or 4 biologically independent samples. Source data are provided as a Source Data file.

## Discussion

Our studies reveal that MYADM binds to PeV-A1 viral particles and is required for viral entry, which points to a role of MYADM as a receptor for human parechoviruses. MYADM was required for PeV-A1, PeV-A2 and PeV-A3 infection of diverse cell types including primary intestinal epithelial cells, which represent a tissue type relevant for parechovirus pathogenesis and fecal-oral transmission. MYADM is essential for integrin-dependent and -independent PeV-A genotypes suggesting a conserved role in parechovirus entry. A recent study independently found MYADM as a host factor for PeV-A3 as well as five other PeV-A genotypes (PeV-A1, 2, 4−6) and showed the interaction between MYADM and VP0 capsid protein of PeV-A3[27]. The identification of a common receptor may lead to the development of antiviral strategies targeting emerging parechoviruses increasingly linked to severe neurological diseases.

Divergence in the amino acid sequence of MYADM between mammalian species controlled the ability to act as host factor for human parechoviruses. Our mutagenesis mapping studies were guided by this diversity and pinpointed specific residues within the fourth extracellular loop critical for receptor function and binding. This suggests that capsid-MYADM interactions could be a host barrier limiting cross-species transmission. Although parechoviruses are frequently isolated from wild animals including rodents and bats, they constitute viral species distinct from human parechoviruses[28,29]. Whether they utilize MYADM or another receptor is unknown. Regardless, the identification of MYADM as species-specific receptor

opens up the possibility to generate mice transgenic for human MYADM to study parechovirus pathogenesis in a small animal model analogous to existing models for other picornaviruses[30,31].

We demonstrated that MYADM binds to PeV-A1 particles and is essential for functional viral entry, which points to a role as receptor. MYADM knockout did not reduce cell binding, suggesting that it does not act as a primary cellular attachment factor for PeV-A1. Rather, binding of PeV-A1 to MYADM during viral entry was enhanced when raising the temperature to allow endocytosis. Moreover, the in vitro binding between MYADM with parechovirus particles was detected at slightly acidic conditions (pH ≤6.5). Whether during a viral infection the binding is similarly dependent on the pH will need to be experimentally validated because in vivo and in vitro binding conditions can differ.

There are intriguing parallels with the receptor usage of the echoviruses, picornaviruses related to the parechoviruses. While different genotypes can use different attachment factors, including CD55 and integrin[10], the neonatal FC receptor (FcRn) was recently identified as pan-echovirus receptor[26,32]. It has a post-attachment function in mediating the uncoating step required for crossing of viral RNA over the membrane[26]. Structural studies have suggested that parechovirus uncoating and formation of the RNA translocation channel proceeds via a mechanism distinct from other, better-studied, picornaviruses[33]. The broad dependence of parechoviruses on MYADM and the binding between parechovirus and MYADM points to an important receptor function post attachment. Whether this function is specifically in

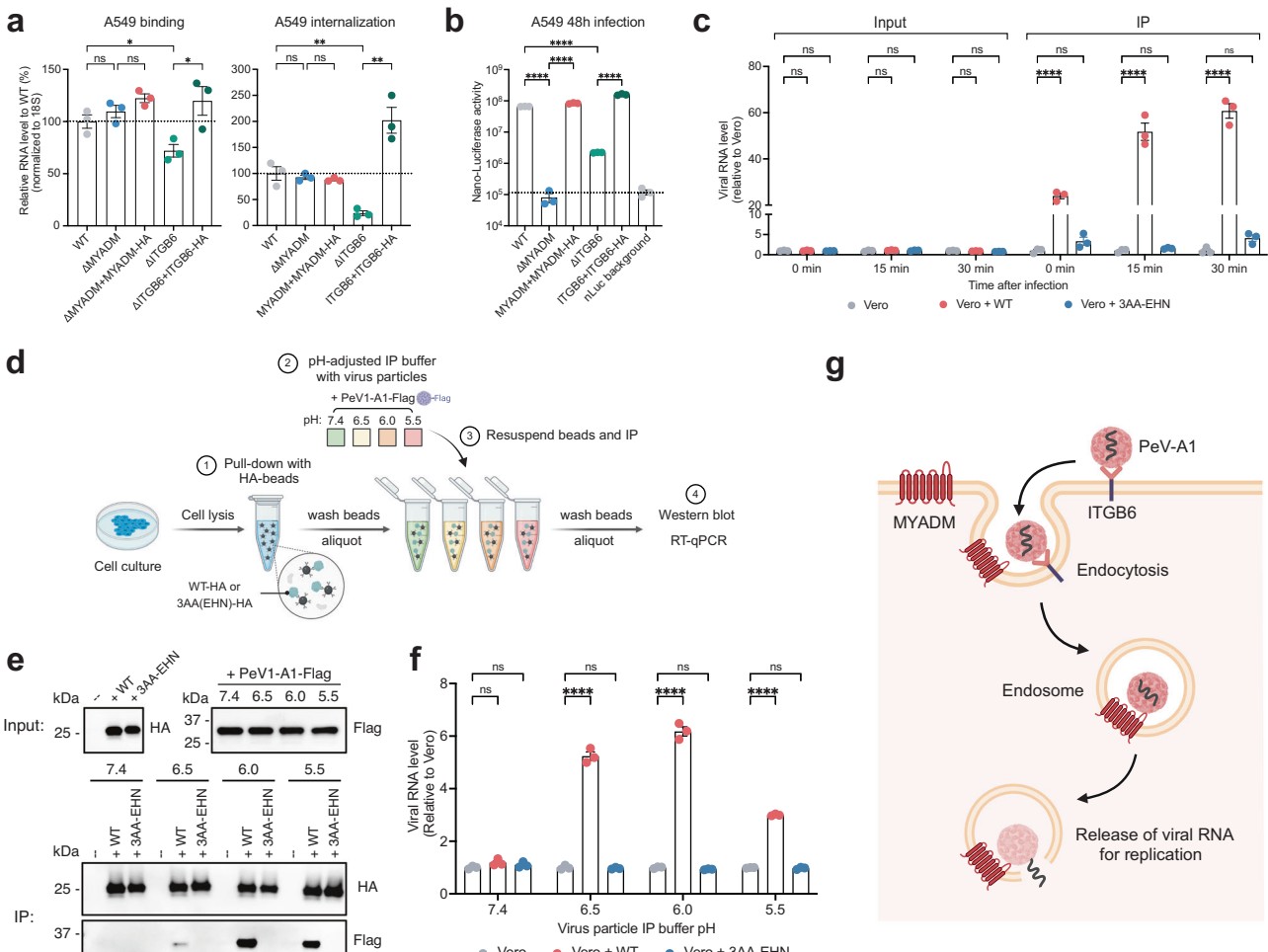

**Fig. 5 | MYADM is essential for parechovirus entry and binds PeV-A1 particles in acidic conditions. a** Wild type (WT), MYADM knockout (ΔMYADM), MYADM-complemented ΔMYADM, ITGB6 knockout (ΔITGB6), and ITGB6-complemented ΔITGB6 A549 cells were subjected to binding and internalization assays with PeV-A1. Viral RNA levels were measured by RT-qPCR and normalized to 18S. Data are presented as mean values +/− SEM, $n = 3$ biologically independent samples, unpaired two-tailed Student's $t$ test. *$P < 0.05$; **$P < 0.01$; ns, not significant. **b** PeV-A1-nLuc infection in A549 cell lines at 48 hpi. PeV-A1-nLuc without cells was used as nLuc background control. Data are presented as mean values +/−SEM, $n = 3$ biologically independent samples, unpaired two-tailed Student's $t$ test, ****$P < 0.0001$. **c** Time point virus particle binding assay using Vero cells, Vero cells expressing HA-tagged human MYADM (Vero + WT), and 3AA-EHN mutant (Vero + 3AA-EHN). Cells were lysed (Input) and immunoprecipitated with anti-HA beads (IP). Viral RNA

levels were measured by RT-qPCR. Data are presented as mean values +/− SEM, $n = 3$ biologically independent samples, two-way ANOVA with Šídák's multiple comparison test, ****$P < 0.0001$; ns not significant. **d** Schematic of the pH-dependent virus particle binding assay. **e** Western blot analysis of HA-tagged MYADM proteins and PeV-A1-Flag virus particles in NETN buffers with varied pH before (Input) and after immunoprecipitation (IP). Data from one experiment representative of three independent experiments ($N = 3$) are shown. **f** Quantification of viral RNA levels after immunoprecipitation using RT-qPCR. Data are presented as mean values +/− SEM, $n = 3$ biologically independent samples, two-way ANOVA with Šídák's multiple comparison test, ****$P < 0.0001$; ns not significant. **g** Model for the role of MYADM in PeV-A1 entry: PeV-A1 primarily binds to cells using ITGB6. MYADM acts as a post attachment receptor through its interaction with virus particles. Source data are provided as a Source Data file.

uncoating, analogous to other picornaviruses, will need to be experimentally addressed in future studies leveraging structural biology approaches such as cryogenic electron microscopy. In particular, structural analysis of parechovirus particles in the presence and absence of MYADM will be informative to test whether the interaction induces conformational changes associated with uncoating.

## Methods

### Cells and viruses

HT-29 (ATCC, HTB-38), heparan-sulfate/sialic acid deficient HT-29 (HT29-DKO, ΔEXTL3 and ΔSLC35A1), A549 (ATCC, CCL-185), 293FT (Thermo Fisher, #R70007), HuTu80 (ATCC, HTB-40), Vero (ATCC, CCL-81), BHK21 (ATCC, CCL-10), and MEF cells were cultured at 37 °C with 5% $CO_2$ in Dulbecco's Modified Eagle medium (DMEM) supplemented with 10% fetal bovine serum (FBS), 2 mM L-glutamine, and 1× penicillin/streptomycin.

Human parechovirus PeV-A1, Harris strain (ATCC, VR-52) and PeV-A3, US/MO-KC/2014/001 (Bei resources, NR-51187) were propagated and titered on Vero cells; PeV-A2, Williamson (ATCC, VR-1820) was propagated and titered on HT29-DKO cells. The GFP-expressing HSV (K26GFP) was kindly provided by Desai and Person[34]. EV-A71, Taiwan/4643/98 strain, was generated from infectious clone using RD cells[35]. Coxsackie Virus B3 Luciferase was a generous gift from Dr. Frank van Kuppeveld.

### Plasmid construction

The complementary DNA (NM_001020818.2) encoding human MYADM protein was codon-optimized and synthesized (IDT). It served as a template for a two-step PCR, where C-terminal HA-tag and vector overhangs complementary to the lentiviral vector were added using the following primers (IDT): 5′-TGTGGTGGAATTCTGCAGATACCATG CCCGTGACGGTCACACGCAC-3′, 5′-TCCGGAACATCATACGGATATAC

CTTAACAAAAACGAGGTGCGCAG-3', and 5'-CGGCCGCCACTGTGCTG GATTTATGCATAATCCGGAACATCATACGGATATACCT-3'. The result-ing cDNA was cloned into the lentivirus vector pLenti-CMV-Puro-Dest (w118-1) (Addgene, 17452) digested with EcoRV using Gibson Assembly. Similarly, the HA-tagged ITGB6 sequence was amplified from integrin beta6 EGFP-N3 (Addgene, 13593) using the primers (IDT): 5'-TGTGG TGGAATTCTGCAGATACCATGGGGATTGAACTGCTTTGCCTG-3', 5'-TC CGGAACATCATACGGATAGCAATCTGTGGAAAGGTCTACCTTTTG-3', and 5'-CGGCCGCCACTGTGCTGGATTTATGCATAATCCGGAACATCAT ACGGATAGCAATC-3'. The amplified sequence was then cloned into EcoRV-digested pLenti-CMV-Puro-Dest (w118-1).

The complementary DNAs encoding HA-tagged MYADM of other mammalian species, including *Mus musculus* (NM_001093764.1), *Equus caballus* (XM_023650340.1), *Canis lupus familiaris* (XM_003638804.3), *Felis catus* (XM_003997448.6), *Mesocricetus auratus* (XM_005084151.4), *Macaca fascicularis* (XM_005590268.3), *Capra hircus* (XM_018063034.1), *Bos taurus* (XM_027514819.1), and *Mustela putorius furo* (XM_013049591.2), were synthesized as gene fragments (IDT and Twist Bioscience). The gene fragments were synthesized with lentiviral vector overhangs, specifically 5'-TGTGGTGGAATTCTGCA-GATACC - MYADM-HA cDNA sequences - ATCCAGCA-CAGTGGCGGCCG-3', and cloned into pLenti-CMV-Puro-Dest (w118-1) as described above. Additionally, ectodomain swapped constructs of the human_MYADM and hamster_MYADM were also synthesized and cloned using similar procedures.

To generate lentiviral constructs expressing the human MYADM with amino acid mutants corresponding to house MYADM, pLenti-CMV-Puro-MYADM-HA was used as a template and mutations were introduced by PCR using primers pairs containing the respective mutations and a 20 bp overlap for Gibson Assembly. The primers used for the mutant constructs are as follows (IDT): 6AA-EHNTHD: 5'-TTC AACATCACATGCCCACTACGTTTGCGACTGGGACCGGCGGCTTGCGG TGGC-3' and 5'-AGTGGGCATGTGATGTTGAACAATTATGTTCCCGTGA TCGTCGGGGTTGCCCTCC-3'; 3AA-EHN: 5'-CGATCACGGGAACATAAT TGTTCAAGATCACATGCCTACTACGTTTGC-3' and 5'-CAATTATGTTC CGTGATCGTCGGGGTTGCCCTC-3'; 3AA-THD: 5'-TTCAACATCACAT GCCCACTACGTTTGCGACTGGGACCGGCGGCTTGCGGTGGC-3' and 5'-AGTGGGCATGTGATGTTGAACAAGATACGTCCCGTGATCGTCGG-3'; 2AA-EH: 5'-TTCAAGATCACATGCCTACTACGTTTGCGCCTGGGACCGG CGGCTTGCGGTGGC-3' and 5'-AGTAGGCATGTGATCTTGAACAAGAAT GTTCCCGTGATCGTCGGGGTTGCCCTCC-3'; 2AA-EN: 5'-TTCAAGATC ACATGCCTACTACGTTTGCGCCTGGGACCGGCGGCTTGCGGTGGC-3' and 5'-AGTAGGCATGTGATCTTGAACAATTTACTTCCCGTGATCGTCG GGGTTGCCCTCC-3'; 2AA-HN: 5'-TTCAAGATCACATGCCTACTACGTT TGCGCCTGGGACCGGCGGCTTGCGGTGGC-3' and 5'-AGTAGGCATGT-GATCTTGAACAATTATGGTCCCGTGATCGTCGGGGTTGCCCTCC-3'; 1AA-E: 5'-TTCAAGATCACATGCCTACTACGTTTGCGCCTGGGACCGGC GGCTTGCGGTGGC-3' and 5'-AGTAGGCATGTGATCTTGAACAAGATAC tTCCCGTGATCGTCGGGGTTGCCCTCC-3'; 1AA-H: 5'-TTCAAGATCACA TGCCTACTACGTTTGCGCCTGGGACCGGCGGCTTGCGGTGGC-3' and 5'-AGTAGGCATGTGATCTTGAACAAGAATGGTCCCGTGATCGTCGGG GTTGCCCTCC-3'; 1AA-N: 5'-TTCAAGATCACATGCCTACTACGTTTGC GCCTGGGACCGGCGGCTTGCGGTGGC-3' and 5'-AGTAGGCATGTGAT CTTGAACAATTTACGTCCCGTGATCGTCGGGGTTGCCCTCC-3'. Like-wise, the human MYADM mutations were introduced into the pLenti-CMV-Puro-horse-MYADM-HA construct by PCR using the following primer pairs (IDT): 3AA-DVS: 5'-CTCAACATCACATGCGCACTATGTA TGCGACTGGGACCGACGGCTCGCTGTGGC-3' and 5'-AGTGCGCATGT-GATGTTGAGCAAGATACGTCCCGAGAGCGGCGCGGCTGTCCGCCG-3'; 3AA-RYA: 5'-CTCAAGATCACATGCGtACTATGTATGCGcCTGGGACCG ACGGCTCGCTGTGGC-3' and 5'-AGTACGCATGTGATcTTGAGCAATT ATGTTCCCGAGAGCGGCGCGGCTGTC-3'; 1AA-V: 5'-CTCAACATCACA TGCGCACTATGTATGCGACTGGGACCGACGGCTCGCTGTGGC-3' and 5'-AGTGCGCATGTGATGTTGAGCAATTTACTTCCCGAGAGCGGCGCG GCTGTCCGCCG-3'.

## Generation of heparan-sulfate/sialic acid deficient HT-29 cell line

The HT29-DKO cell line was generated through a sequential knockout of SLC35A1 and EXTL3 genes of HT-29 (ATCC, HTB-38) cells. To obtain single clonal SLC35A1 knockout HT-29 cells, the pX458 vector (Addgene, #48138) expressing Cas9 and small guide RNA (sgRNA) against SLC35A1 (SLC35A1-sgRNA: TATAACTTCTGTGATACACA) was utilized. GFP-positive single cells were sorted 48 h post-transfection using BD Aria II into 96-well plates and screened for knockout by Sanger sequencing using primers 5'-TGCACGCTCATGTAATTGGC-3' and 5'-GTAGCATCTCTCAAGTTGTGAC-3' (IDT). To generate SLC35A1 and EXTL3 double knockout cells, SLC35A1 knockout cells were transfected with pX458 vector encoding sgRNA against EXTL3 (EXTL3-sgRNA, AAATGAACCTCGGTAACACG). Similarly, GFP-positive cells were single-cell sorted and screened for knockout using Sanger sequencing with primers 5'-GCTGAAGCTCTCCACCTTCG-3' and 5'-AGGTATCTGGGTGAGGCGTAG-3' (IDT). The selected clone was fur-ther validated through Western blot analysis using an EXTL3-specific antibody (1:1000, Santa Cruz Biotechnology, clone G-5, sc-271986) and immunofluorescence staining using Rhodamine labeled Peanut agglutinin (Vector Laboratories, RL-1072) and an antibody against heparan sulfate (Amsbio, clone F58-10E4, 370255-S).

## CRISPR-Cas9 screen and data analysis

The CRISPR screens were performed using the heparan-sulfate/sialic acid deficient HT-29 cell line as previously described[36]. In brief, HT29-DKO cells were stably transduced with lentiCas9-Blast (Addgene, #52962) and selected using blasticidin. Subsequently, HT29-DKO-Cas9 cells were transduced with packaged sgRNA lentivirus of the Brunello library at an MOI of 0.3. After selection with puromycin, cells were pooled and expanded. Approximately 100 million mutagenized cells, constituting over 1000× sgRNA coverage, were inoculated with PeV-A1 and PeV-A2 at MOI of 0.1 and incubated until over 95% of the cells were dead. The medium was changed and the remaining live cells grew to form colonies. The unselected starting population was used as an uninfected reference. Genomic DNA was extracted from the control cells and virus-resistant cells using QIAamp DNA Mini Kit (Qiagen). The sgRNAs were amplified and subjected to next-generation sequencing using an Illumina HiSeq Lane via Novogene. The sgRNA sequences were analyzed with the MAGeCK algorithm[37]. The FASTQ files for CRISPR screens generated in this study have been deposited in ArrayExpress under accession code E-MTAB-13894. The Mageck ana-lysis of the CRISPR screens are provided as Supplementary Data 1.

## Generation of CRISPR-Cas9-mediated knockout cell lines

Guide RNA sequences for CRISPR editing were extracted from the Brunello library and synthesized with IDT: MYADM-sgRNA (CTCCAC-GATGAGGATGATCA) and ITGB6-sgRNA (CCAGACTGAGGACTACCC GG) were cloned into pX458 (Addgene, #48138) to generate clonal knockout cell lines. Cells were transfected with pX458 constructs using Mirus TransLT1 Transfection Reagent (Mirus, MIR2300) and single-cell sorted based on GFP expression into 96-well plates using the BD influx cell sorter two days post transfection. Clones were expanded and screened by sequencing analysis. Genotyping was performed by extracting genomic DNA from the cells using QuickExtract DNA Extraction Solution (Lucigen), PCR amplifying a 500-700-base pair (bp) region encompassing the sgRNA-targeted site, and then Sanger sequencing the amplicons with a nested sequencing primer. The sequencing results were analyzed using the SYNTHEGO ICE analysis tool.

The primer sequences used for genotyping were as follows (IDT): MYADM-F: 5'-GCACCACCATCACAACCACC-3' and MYADM-R: 5'-GCTA GGATGAAGCAGATGGCGTAC-3', MYADM-seq: 5'-CAGCTGGTGTCTA CCTGC-3'; ITGB6-F: 5'-CCTCCAATAGTTCTAAACTCACTG-3' and ITG B6-R: 5'-CCTCACCTCCACTCTAAGCAAC-3', ITGB6-seq: 5'-GCCTCTGA-GAGAACTGCTCG-3'.

## Lentivirus production and transduction

Heterogeneous stable cell lines for gene expression or knockout were generated using lentiviral transduction. Lentivirus stocks were produced in 293FT cells by co-transfection of the interested plasmid and packaging plasmids ΔVPR, VSV-G and pAdvantage using TransIT-LT1 Transfection Reagent. One day post transfection, the transfection medium was replaced by pre-warmed DMEM culture medium. The medium containing lentivirus was harvested at 24 h and 48 h, filtered using 0.45-micron syringe filters, and stored at −80 °C.

For lentiviral transduction, target cells were seeded at $3 \times 10^5$ cells/well in a 6-well plate, with an addition well as a no transduction control. The lentivirus solution (2–4 ml per well) was added the following day, supplemented with protamine sulfate at a final concentration of 8 µg/mL, and cells were incubated at 37 °C. Approximately 48 h post transduction, cells were trypsinized, transferred to a T75 flask, and selected with puromycin at the following concentrations: HT29-DKO (1 µg/mL), HuTu80 (0.4 µg/mL), 293FT (2 µg/mL), A549 (1.5 µg/mL), MEF (5 µg/mL), Vero (6 µg/mL).

## Construction of PeV-A1 infectious clone and reporter virus constructs

HT29-DKO cells were seeded in a 6-well plate and infected with PeV-A1. Viral RNA was extracted from the culture medium of virus-infected cells using QIAamp Viral RNA Mini Kit (Qiagen) and reverse transcribed with SuperScript III First-Strand Synthesis System (Invitrogen). Three overlapping fragments covering the full-length PeV-A1 genome were amplified from cDNA, and a backbone containing a T7 promoter was PCR-amplified from pD2/IC-30P[38]. These four fragments (F1-F4) were assembled using the Gibson Assembly Master Mix (New England Biolabs). The clones were screened by colony PCR and confirmed by whole-plasmid sequencing (Primordium Labs). Primers used for viral genome and backbone amplification were as follows (IDT): PeV-A1-F1: 5′-TTAATACGACTCACTATAGGTTTGAAAGGGGTCTCCTAGAGAGCTTG-3′ and 5′-AGTGACAACTGGTTTAGGTTCATATACTACC-3′; PeV-A1-F2: 5′-AACCTAAACCAGTTGTCACTTATGATTCAAAACTG-3′ and 5′-TCTAGTTGGTACACATGGTCTCCAAC-3′; PeV-A1-F3: 5′-GACCATGTG-TACCAACTAGATTCAGATGAC-3′ and 5′-TCGATAAGCTTGCCTGCAGGTTTTTTTTTTTTTTTTTTTTTTTTTTTTTTTTGTCATGTCCAATGTTCCAAGTTAGTGTC-3′; F4 (backbone): 5′-CCTGCAGGCAAGCTTATCGATGA-TAAG-3′ and 5′-CCTATAGTGAGTCGTATTAAATTTACGCGTGAG-3′.

The PeV-A1 infectious clone, designated pD2/IC-PeV-A1, was used to generate reporter virus constructs. A mNeon or NanoLuc reporter gene and a porcine teschovirus (P2A) cleavage site were introduced between PeV-A1 5′ UTR and the start codon of the polyprotein. The reporter gene and partial viral genome were synthesized (IDT) and inserted into pD2/IC-PeV-A1 between the KpnI and AvrII restriction enzyme sites using Gibson Assembly.

To produce PeV-A1-mNeon/nLuc reporter viruses, the constructs were linearized with SbfI and in vitro transcribed with MEGAscript T7 Transcription Kit (Invitrogen). The viral RNA was then subjected to DNase digestion and purified using Lithium Chloride precipitation. Vero cells ($2 \times 10^6$ cells) were washed twice with PBS, mixed with purified viral RNA (5 µg) in electroporation buffer (Teknova, E0399), and electroporated using Bio-Rad Gene Pulser Xcell electroporator with the square wave protocol. The electroporated cells were resuspended in cell culture medium and transferred into a T175 flask for virus propagation. Both cells and supernatant were harvested when over 80% cell death was observed and separated by centrifugation at $600 \times g$ for 10 min at 4 °C. The cell pellets were subjected to three freeze-thaw cycles to disrupt the cells, followed by centrifugation to remove cellular debris. The resulting supernatant was combined with the original supernatant, filtered through a 0.22 µm filter, and stored at −80 °C as aliquots.

## Quantification of virus infection in cell lines

**qPCR infectivity assays.** Cells were plated in 96-well plates in triplicates and infected with viruses at an MOI of 0.1. Cells were collected and processed at 48 h post infection using Power SYBR Green Cells-to-$C_T$ Kit (ThermoFisher, #4402954) following the manufacturer's instructions. All samples were normalized to 18S expression. The following qPCR primers were used (IDT): PeV-A1-3-F: 5′-ACGAAGGAT GCCCAGAAGGTAC-3′, PeV-A1-3-R: 5′-AGGGATCCCCCCTGGGTTTG-3′; 18S-F: 5′-AGAAACGGCTACCACATCCA-3′, 18S-R: 5′-CACCAGACTTGCC CTCCA-3′.

**Luciferase reporter virus assays.** Cells were seeded in triplicates in 96-well plates, infected with reporter virus at an MOI of 0.1 or 1, and incubated at 37 °C with 5% CO2. Luciferase activity was measured at the indicated time points using Nano-Glo Luciferase Assay System (Promega, N1120) or Renilla Luciferase Assay System (Promega, E2810), and immediate luciferase readings were taken using a Glomax luminometer.

**GFP reporter virus incucyte analysis.** Cells were seeded in 96-well plates in triplicates and infected with PeV-A1-GFP reporter virus at MOI of 0.1 or 1. Individual wells were imaged over time at 4× magnification using an Incucyte System (Sartorius) in a 37 °C, 5% CO2 incubator. Virus infection was tracked by GFP fluorescence, and the number of GFP-positive foci per image was counted using Incucyte Analysis Software.

## Crystal violet staining

HT29-DKO cell lines were seeded in a 24-well plate one day before infection. The following day, cells were infected with PeV-A1 at an MOI of 0.5 and incubated at 37 °C. After 72 h, cells were washed with 1× PBS, fixed with 4% PFA in 1× PBS, and stained with crystal violet staining solution (0.5% crystal violet, 20% methanol in $H_2O$).

## Construction of PeV-A1 replicon and replicon assays

The PeV-A1 replicon was constructed by deleting a total of 1161 bp from the VP3 and VP1 sequences. The PeV-A1-nLuc served as a template and was PCR-amplified into two fragments (F1 and F2) with 20 bp overlap at both ends by Phusion High-Fidelity DNA Polymerase (ThermoFisher, F530L) and the following primer pairs (IDT): F1 (4522 bp, 5′-CTGGTGCTCAATCAGTGGCTCCCTTCTTGACTGGAACCTCTACTGG-3′ and 5′-TCGATAAGCTTGCCTGCAGGTTTTTTTTTTTTTTTTTTTTTTTTT TTTTTTTGTCATGTCCAATGTTCCAAGTTAGTGTC-3′) and F2 (5332 bp, 5′-CCTGCAGGCAAGCTTATCGATGATAAG-3′ and 5′-AGCCACTGATT-GAGCACCAGTTGTACAAAG-3′). The fragments were assembled using Gibson Assembly, and the clones were confirmed by whole-plasmid sequencing.

To generate replicon RNA, the construct was linearized with SbfI, in vitro transcribed using MEGAscript T7 Transcription Kit (Invitrogen), and purified using Lithium Chloride precipitation. Two million cells were washed twice with PBS and resuspended in electroporation buffer (Teknova, E0399) containing 5 µg of replicon RNA. The mixture was transferred into a Gene Pulser Electroporation Cuvette (Bio-rad, 1652089), and electroporation was performed using the square wave protocol. The electroporated cells were resuspended in cell culture medium and aliquoted at $2 \times 10^4$ cells per well into a 96-well plate, with three replicates for each cell line. Cells were lysed at the indicated time points, and luciferase activity was measured using Nano-Glo Luciferase Assay System (Promega, N1120) and a Glomax luminometer.

## Virus binding and internalization assays

The assays were conducted following the methods described previously with slight modification[39]. Briefly, A549 cell lines were plated in 12-well plates with three replicates each, one day prior to infection. PeV-A1 (MOI of 1) was added to the pre-chilled cells and incubated on

ice for 1 h. Unbound virus particles were removed by washing the cells 6 times with ice-cold PBS supplemented with 2% bovine serum albumin (BSA). For virus binding assay, cells were lysed directly in the plate after the washes by adding 350 μl RLT plus buffer with β-mercaptoethanol and collected for RNA extraction using a RNeasy Plus Mini Kit (Qiagen). For internalization assay, after 6 washes with ice-cold PBS and 2% BSA, pre-warmed DMEM medium supplemented with 2% FBS was added to cells. Cells were incubated at 37 °C for 1 h for virus internalization. Cells were then chilled on ice and incubated with 500 ng/mL proteinase K in PBS at 4 °C for 2 h to remove residual surface-bound virus particles. Cells were lysed and collected for RNA extraction after 6 additional washes with ice-cold PBS and 2% BSA as described above. The qRT-PCR was performed using iScript™ Reverse Transcription Supermix (Bio-Rad) and SsoAdvanced Universal SYBR Green Supermix (Bio-Rad) following the manufacturer's manual. The same qPCR primer sets of PeV-A1-3 and 18S were used as those listed in the qPCR infectivity assay.

### Western blot analysis

To detect the endogenous MYADM, cells were collected by scraping from 10-cm dishes and lysed in 500 μL lysis buffer (25 mM Tris-HCl, pH 7.5, 150 mM NaCl, 5 mM EDTA, 1% Triton X-100, and 1× Halt protease inhibitor). The lysates were passed through a 22 G needle and the insoluble materials containing the detergent-resistant membranes (DRMs) were collected by centrifugation at full speed for 15 min at 4 °C. The pellets were resuspended in 40 μL of lysis buffer supplemented with 60 mM octyl-glucoside and incubated at 37 °C for 30 min to solubilize the DRMs. The supernatants were collected by centrifugation and mixed with 4× Laemmli Sample Buffer (Bio-rad, 1610747) supplemented with β-mercaptoethanol for Western blot analysis. Protein samples were then separated by SDS-PAGE on 4–15% Mini-PROTEAN TGX Precast Gels (Bio-Rad, 4561084) and transferred onto PVDF membranes. PVDF membranes were blocked for 1 h in 5% non-fat milk in PBS containing 0.1% Tween-20 (PBST). Subsequently, membranes were incubated at 4 °C overnight with MYADM antibody (1:500; mAb 2B12)[13] diluted in blocking buffer. After washing three times with PBST, membranes were incubated with goat anti-mouse IgG antibody (1:5000; GeneTex, GTX213111-01) for 1 h at room temperature. For loading controls, membranes were incubated with HRP-conjugated GAPDH antibody (1:5000; GTX627408-01) for 1 h at room temperature. Membranes were subjected to SuperSignal West Pico PLUS Chemiluminescent Substrate (ThermoFisher, 34580) after PBST washes and visualized with a Bio-Rad ChemiDoc Touch Imaging System.

To detect the expression of HA-tagged MYADM proteins in lentiviral transduced cell lines, two million cells were lysed with 300 μL lysis buffer (100 mM Tris-HCl, pH 8.0 and 1% SDS) and incubated at 98 °C for 15 min. HA Tag Polyclonal Antibody (1:1000; ThermoFisher, 14-6756-81) and goat anti-rabbit IgG antibody (1:5000; GeneTex, GTX213110-01) were used for detection.

### Generation of Flag-tagged PeV-A1 virus particles

The Flag-tagged PeV-A1 virus construct was generated by using pD2/IC-PeV-A1 as the cloning template. Through PCR, the virus construct was fragmented into three segments with at least a 20 bp overlap, while also introducing the Flag-tag sequence into the structural protein VP1. The primers used were as follows (IDT): 5′-GATTACAAG GATGACGACGATAAGAACCTTACAAATCAGAGTCCATATGGTCAAC-3′ and 5′-TCGATAAGCTTGCCTGCAGGtttttttttttttttttttttttttttttGTCATG TCCAATGTTCCAAGTTAGTGTC-3′; 5′-TTAATACGACTCACTATAGGTT TGAAAGGGGTCTCCTAGAGAGCTTG-3′ and 5′-CTTATCGTCGTCATCC TTGTAATCTGCCATATCACCCCGTAGTGCACGAC-3′; 5′-CCTGCAGGC AAGCTTATCGATGATAAG-3′ and 5′-CCTATAGTGAGTCGTATTAAATT TACGCGTGAG-3′. The three fragments were then assembled by Gibson Assembly.

PeV-A1-Flag viruses were propagated using the same procedures outlined to produce the mNeon/nLuc reporter viruses as described above. To validate the infectivity of the Flag-tagged PeV-A1 virus particles, the PeV-A1-Flag viruses were passaged three additional times on Vero cells. The infected cells were collected for RT-PCR and Western blot analysis. For RT-PCR, RNA was extracted from Vero cells using the RNeasy Plus Mini Kit (Qiagen) and then reverse transcribed into cDNA using iScript™ Reverse Transcription Supermix (Bio-Rad). To assess the stability of the Flag-tag insertion, primers that flank the Flag-tag region were utilized (IDT): 5′-TGGTTATTTGATGTGCAAGCCCTTC-3′ and 5′-TCTAGTTGGTACACATGGTCTCCAAC-3′, which yield a fragment size of 294 bp or 270 bp with or without the Flag-tag insertion.

### Virus particles co-immunoprecipitation

**Virus propagation and concentration.** Vero cells were infected with either PeV-A1 or PeV-A1-Flag and RD cells were infected with EV-A71 at a MOI of 0.1. After 2 days post infection, both cells and supernatant were harvested and separated by centrifugation at $600 \times g$ for 10 min at 4 °C. The cell pellets were then subjected to three freeze-thaw cycles, followed by centrifugation to remove any cellular debris. The resulting supernatant was combined with the original supernatant, filtered through a 0.22 μm filter, and subjected to ultracentrifugation at $320,000 \times g$ for 4 h at 4 °C over a sucrose cushion. The pellets containing the viral particles were resuspended in PBS, and the insoluble material was removed by centrifugation at $16,000 \times g$ for 10 min at 4 °C. The supernatant was further clarified with a Zeba Spin Desalting Column (Thermo Fisher Scientific).

**pH dependent binding assay.** Fifteen million cells per cell line (Vero cells, Vero cells expressing human MYADM-HA or MYADM-3AA-EHN-HA) were seeded in a 15-cm dish and lysed in ice-cold lysis buffer (25 mM Tris-HCl, pH 7.5, 150 mM NaCl, 5 mM EDTA, 1% Triton X-100, 60 mM octyl-glucoside, and 1× Halt protease inhibitor). The cells were disrupted using a Dounce homogenizer and incubated on ice for 30 min. The lysates were centrifuged at $16,000 \times g$ for 20 min at 4 °C and the resulting supernatant was subjected to immunoprecipitation using Pierce Anti-HA Magnetic Beads (ThermoFisher, 88836). Beads were incubated with the lysates for 3 h at 4 °C, washed three times with the lysis buffer, and resuspended in 800 μL PBS.

To test potential binding of MYADM with concentrated virus particles, virus was added to NETN buffer (50 mM Tris-HCl, 150 mM NaCl, 1 mM EDTA, 0.5% NP-40) adjusted to pH 7.4, 6.5, 6.0 or 5.5 and supplemented with 1× Halt protease inhibitor. The HA beads were aliquoted into four tubes, resuspended in NETN buffer containing approximately $2 \times 10^7$ PFU virus at the respective pH values, and incubated at 4 °C for 2 h. After incubation beads were washed with NETN buffer adjusted to the respective pH. One third of the beads was subjected for RNA extraction using Trizol reagent (Invitrogen, 15596018) following the manufacturer's protocol. The RT-qPCR was performed using iScript™ Reverse Transcription Supermix (Bio-Rad) and SsoAdvanced Universal SYBR Green Supermix (Bio-Rad) using the primers listed above. The remaining beads were resuspended in 4× Laemmli Sample Buffer (Bio-rad, 1610737) supplemented with β-mercaptoethanol and incubated at 98 °C for 5 min for Western blot analysis. HA Tag Polyclonal Antibody (1:1000; ThermoFisher, 14-6756-81), Flag Tag Antibody (1:1000; Cell Signaling Technology, 2368S) and goat anti-rabbit IgG antibody (1:5000; GeneTex, GTX213110-01) were used for detection.

**Time point binding assay.** Six million cells were plated in a 10-cm dish for each condition and infected with purified PeV-A1 or EV-A71 at an MOI of 1. The infection was carried out on ice for 1 h followed by washing three times with ice-cold PBS. The cells were further incubated at 37 °C for 0, 15 or 30 min. After three subsequent washes with

PBS, the cells were lysed using ice-cold lysis buffer (25 mM Tris-HCl, pH 7.5, 150 mM NaCl, 5 mM EDTA, 1% Triton X-100, 60 mM octyl-glucoside, and 1× Halt protease inhibitor) and dounce lysed. The lysates were then centrifuged, and the soluble fractions were subjected to immunoprecipitation using anti-HA magnetic beads. The beads were washed with the lysis buffer and subsequently resuspended in Trizol reagent for RNA extraction and then qRT-PCR as described above.

## Protein alignment and phylogenetic analysis

The full length MYADM protein sequences of various mammalian species, *Homo sapiens* (NP_001018654.1), *Mus musculus* (NP_001087233.1), *Equus caballus* (XP_023506108.1), *Canis lupus familiaris* (XP_003638852.1), *Felis catus* (XP_003997497.1), *Mesocricetus auratus* (XP_005084208.1), *Macaca fascicularis* (XP_005590325.1), *Capra hircus* (XP_017918523.1), *Bos taurus* (XP_027370620.1), and *Mustela putorius furo* (XP_012905045.1), were used for the alignment (https://www.genome.jp/tools-bin/clustalw) and for the construction of the cladogram (http://phylogeny.lirmm.fr/phylo_cgi/index.cgi)[40].

## Fluorescent In Situ Hybridization (FISH) assay

Cells were seeded on 8-well chamber slides. For virus binding assay, cells were incubated on ice with PeV-A1 (MOI 100) for 1 h, and the unbound virus was removed by washing with cold PBS. For virus internalization, cells were then incubated with pre-warmed media for 30 min at 37 °C. FISH was performed using the ViewRNA ISH Cell Plus Assay (ThermoFisher) according to the manufacturer's instructions. PeV-A1 RNA was detected using customized PeV-A1 specific probes (Alexa Fluor 546, ThermoFisher). Following RNA labeling, cells were stained with Wheat Germ Agglutinin (Alexa Fluor 488, Invitrogen) and Hoechst for 10 min at room temperature. Subsequently, cells were washed with PBS and mounted using Vectashield mounting medium (Vector Laboratories, H-1000). Images were acquired with an inverted confocal microscope (Zeiss LSM 800) and processed with ZEN software (Zeiss). Image analysis was performed with the FIJI software.

## Human colonoid cultivation

Colon organoids were derived from de-identified, surgically obtained human gastrointestinal tissue as described previously[22] by the lab of Calvin Kuo at Stanford University. Tissues were obtained through the Stanford Tissue Bank with patient consent and approval from the Stanford University Institutional Review Board. Age and sex of the patients were not collected, and samples were taken without a particular targeted or planned enrollment[22].

Colonoids were maintained and passaged as previously described with slight modification[22]. In brief, colonoids were embedded in Cultrex Reduced Growth Factor Basement Membrane Matrix, Type II (BME) and seeded into a 24-well tissue culture plate in a 30 μL droplet per well. BME was polymerized by incubating at 37 °C for 10 min. Then, 400 μL growth medium, which contains Advanced Dulbecco's modified Eagle medium/F12, 50 ng/mL EGF, 10 mM nicotinamide, 10 nM Gastrin, 500 nM A83-01, 10 μM SB202190, 1× B27, 1 mM HEPES, 1 mM N-acetyl-cysteine, 1× Glutamax, and 50% L-WRN-conditioned medium (contains Wnt3a, R-spondin 3 and Noggin), supplemented with 10 μM Y-27632 was added on top of the BME. L-WRN-conditioned medium was generated from L-WRN cells (ATCC, CRL-3276) following a previous description[41]. Growth media was replaced every 2–4 days. Colonoids were passaged every 6–10 days by digesting in TrypLE Express for 10 min at 37 °C and dissociated into single cells by pipetting. Cells were counted, reseeded into fresh BME at a concentration of $2.0 \times 10^4$ cells per well, and plated in 24-well plates.

## Generation of human colonoid MYADM knockout lines

Gene knockout in human colonoid was achieved using the CRISPR-Cas9 system, following a published protocol[42] with some modifications. Briefly, MYADM-sgRNA was cloned into lentiCRISPRv2

(Addgene, #52961), and the plasmid was used for lentivirus production. Lentivirus solution was collected and concentrated using PEG-*it*™ Virus Precipitation Solution (System Biosciences) following the manual. The virus pellet was then resuspended in 600 μL CMGF(-) medium that consists of Advanced DMEM/F12 medium supplemented with 100 U/mL penicillin-streptomycin, 10 mM HEPES buffer and 1× Glutamax and stored at −80 °C.

**Lentivirus transduction.** Colonoid cultures were maintained and passaged in BME matrix as described above. In a 24-well plate, one well of colonoid was digested in TrypLE Express for 5 min at 37 °C, and then CMGF(-) medium with 10% FBS was added to neutralize the trypsin. The digested colonoids were fully dispersed into single cells by pipetting 50 times. The cells were washed by CMGF(-) medium, aliquoted into two tubes, and pelleted for lentiviral transduction. Lentiviral mixture containing 300 μl of concentrated lentivirus, 10 μM Y-27632, and 8 μL of polybrene (10 mM stock) was prepared and transferred to a tube containing the digested colonoids. For the control tube, 300 μL of CMGF(-) with Y-27632 and polybrene was added. Cells were resuspended by pipetting and then transferred from each tube to a well of a 48-well plate. The 48-well plate was incubated at 37 °C for 5 min, sealed with parafilm, and spin-inoculated at 300 g for 1 h at room temperature. The supernatant was then carefully removed and the cell layer in each well was resuspended with 500 μL of ice-cold CMGF(-) medium and transferred to a new microtube. Cells were pelleted and embedded into BME matrix and seeded into three wells in a 24-well plate. The BME matrix was solidified by incubating at 37 °C for 10 min, and growth medium containing 10 μM Y-27632 was added to each well.

**Recovery and selection.** After transduction, colonoids were allowed to recover for 6 days by changing the growth medium every other day, and then selected with 2 μg/mL puromycin. Colonoids were passaged every 7–10 days with the selection for three weeks. After the selection, the colonoids were passaged and expanded into multiple wells without puromycin.

**Single-cell cloning.** Colonoids were digested with TrypLE Express following the same procedures used during transduction. Additionally, the cells were filtered twice through a 70 μm cell strainer to retain single cells. On day 2 after passaging, one well of colonoids was collected in ice-cold 1× PBS and resuspended in 30 μL BME. Clonal MYADM knockout colonoid lines were obtained by limiting dilution. To ensure proper dilution, three additional 1:50 serial dilutions of the BME cell suspensions were tested by inoculating 3 μL per well into a 96-well plate. The dilution resulting in a single colonoid per well was used to plate four 96-well plates. Single clones were visually examined using bright-field microscopy, labeled, and medium was changed every 3 days.

**Single colonoid clone passaging and genotyping.** Single-cell-derived colonoids were passaged after 2–3 weeks using TrypLE Express. Each dissociated clones were resuspended with 60 μL of BME and inoculated into four wells in a 48-well plate. After 1–2 weeks of incubation, one well of each clone was sacrificed for validation. Cells were collected as described for passaging but were resuspended in QuickExtract DNA Extraction Solution for genomic DNA extraction. The genotyping method is referring to that used for the clonal knockout cell lines.

## Virus infection in human colonoids

After 7–10 days of growth colonoids were removed from the BME matrix for polarity reversal to expose the apical surfaces[22]. Matrix-embedded colonoids were dislodged from the 24-well plate by adding 500 μL ice-cold 5 mM EDTA in PBS to each well. The number of wells

was determined by dividing the dissociated colonoids from one well into three wells for experimental treatment. The solution from up to four wells was transferred into a 15-mL conical tube containing 10–12 mL of ice-cold 5 mM EDTA in PBS. The conical tubes were incubated on a rotary shaker at 4 °C for 1 h. Subsequently, colonoids were pelleted by centrifuging in a swinging bucket rotor at $300 \times g$ for 3 min at 4 °C, washed with 5 mL DMEM.

**Virus infection of human colonoids.** Colonoids were resuspended in growth medium containing 10 µM Y-27632 with or without 10% BME matrigel for apical-out or basal-out colonoids. The suspension culture was aliquoted with 400 µL per well into 24-well ultra-low-attachment plates, and incubated for 1 day in a 37 °C, 5% $CO_2$ incubator.

Colonoids were infected with approximately $5 \times 10^4$ PFU of PeV-A1, PeV-A2, and K26GFP per well. The inoculum was removed after 1 h post infection by transferring colonoids in suspension to 1.5 mL LoBind microcentrifuge tubes. The colonoids were collected by centrifuging at $300 \times g$ for 3 min at 4 °C, washed once with DMEM, and transferred to a new 24-well ultra-low-attachment plate in 400 µL of growth medium with 10 µM Y-27632.

**Immunofluorescence.** Colonoids were collected at 48 h post infection, fixed in 2% paraformaldehyde fixative solution for 30 min at room temperature, and washed with PBS. For PeV-A1 infected samples, colonoids were incubated with Anti-dsRNA Antibody, clone rJ2 (1:100; Sigma-Aldrich, MABE1134) in blocking/permeabilization buffer (3% bovine serum albumin, 1% saponin, 1% Triton X-100 and 0.02% sodium azide in PBS) overnight at room temperature and washed with PBS. Subsequently, the colonoids were incubated with Alexa Fluor 488 Goat anti-Mouse secondary antibody (1:500; ThermoFisher, A-11001), Alexa Fluor 594 phalloidin (ThermoFisher, A12381), and 4′,6-diamidino-2-phenylindole dihydrochloride (DAPI; 1:500) in blocking/permeabilization buffer for 4 h at room temperature. In the case of K26GFP infected colonoids, they were incubated with Alexa Fluor 488 anti-GFP polyclonal antibody (ThermoFisher, A-21311), Alexa Fluor 594 phalloidin, and DAPI in blocking/permeabilization buffer for 4 h at room temperature. Stained colonoids were washed with PBS and mounted onto glass slides using Vectashield mounting medium (Vector Laboratories, H-1000), and glass coverslips were affixed using vacuum grease. Organoids were imaged on a LSM 700 confocal microscope (Carl Zeiss) at a 20x magnification.

**qPCR quantification.** Colonoids were collected by centrifuging at the indicated time points. RNA extraction was performed using the RNeasy Micro Kit (Qiagen, 74004), and reverse transcription was carried out using iScript™ Reverse Transcription Supermix (Bio-Rad) following the manufacturer's instructions. qPCR was performed using SsoAdvanced Universal SYBR Green Supermix (Bio-Rad) as described above using the qPCR primer pairs of PeV-A1-3, 18S, and HSV (HSV-UL44-F: 5′-GAGGAGGTCCTGACGAACATCACC-3′, HSV-UL44-R: 5′-CCGGTGACAGAATACAACGGAGG-3′).

**Plaque assay.** Growth medium from virus-infected samples at the indicated hours post infection was collected by centrifugation. Vero cell monolayers were seeded over 80% confluency in 24-well plates one day before infection. Serially diluted virus supernatants were added to cells and incubated for 1 h at 37 °C in 5% $CO_2$. The inoculum was then removed, and the cells were overlaid with 3 ml of 1.2% Avicel/DMEM overlay medium per well. The plates were further incubated at 37 °C in 5% $CO_2$. The overlays were removed by aspiration, and cells were washed with PBS. Cells were then fixed with 4% paraformaldehyde in PBS for at least 30 min and stained with crystal violet staining solution.

## Statistics and reproducibility

One-way or two-way analysis of variance (ANOVA) with Sidak's multiple comparison tests, or two-sided unpaired t tests, were used to assess statistical differences. All statistical analyses were carried out in GraphPad Prism 10 software. Specific statistical tests applied to individual data sets are specified in the corresponding figure legends. The number of individual biological replicates for all graphical representations are shown in the figure legends. All statistical analyses and exact $P$-values are included within the Source Data file.

## Reporting summary

Further information on research design is available in the Nature Portfolio Reporting Summary linked to this article.

## Data availability

The FASTQ files for CRISPR screens generated in this study have been deposited in ArrayExpress under accession code E-MTAB-13894. The Mageck analysis of the CRISPR screens are provided as Supplementary Data 1. MYADM protein sequences: *Homo sapiens* (NP_001018654.1), *Mus musculus* (NP_001087233.1), *Equus caballus* (XP_023506108.1), *Canis lupus familiaris* (XP_003638852.1), *Felis catus* (XP_003997497.1), *Mesocricetus auratus* (XP_005084208.1), *Macaca fascicularis* (XP_005590325.1), *Capra hircus* (XP_017918523.1), *Bos taurus* (XP_027370620.1), and *Mustela putorius furo* (XP_012905045.1), are available from NCBI database. Datasets analyzed during the current study are appended as supplementary data. Source data are provided with this paper.

## Material availability

All unique biological material is available upon request to the corresponding author.

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

## Acknowledgements

The authors thank the Carette lab members for intellectual discussions and support. We thank Calvin Kuo (Stanford University) for the generation of the patient-derived colonoid cell line, Prashant Desai (Johns Hopkins University) for generously providing the GFP-expressing HSV (K26GFP), Manuel Amieva (Stanford University) for advice and oversight on the generation of colonoids, and Denise Monack (Stanford University) for access to their confocal microscope. We acknowledge the valuable assistance of Miguel A. Alonso (Centro de Biología Molecular Severo Ochoa) in providing the anti-MYADM mAb and the detection protocol for endogenous MYADM. Some illustrations were created with BioRender.com. This work was funded in part by the National Institutes of Health (NIH) R01 AI153169 (J.E.C.), NIH R01 AI169467 (J.E.C.), Burroughs Wellcome Fund Investigators in the Pathogenesis of Infectious Disease (J.E.C.), Stanford Maternal & Child Health Research Institute Postdoctoral Support Funds (W.Q.), NIH R01 AI125249 (H.B.G.), and VA Merit Grant (GRH0022) (H.B.G.).

## Author contributions

W.Q. conceived, designed and executed experiments, analyzed data, and prepared the manuscript. C.M.R. built the HT29-DKO mutagenized library, performed the CRISPR screens with PeV-A1 and PeV-A2, and generated the HT29-DKO MYADM knockout cell line. Y.K. provided the wild type human colonoid cell line and guidance for colonoid culture. J.R.Z. analyzed the incucyte data. S.D. and H.B.G. provided the HT29-DKO cells. J.E.C. designed and supervised the research, interpreted the data, and prepared the manuscript. All authors provided comments for the manuscript before submission.

## Competing interests

The authors declare no competing interests.
