## [Peer Review File · Nature Communications]

REVIEWER COMMENTS

Reviewer #1 (Remarks to the Author):

Qiao et al. used a genome-wide CRISPR-Cas9 screen to identify MYADM as an essential host factor involved in paraechovirus (PeV-A) entry. Genetic depletion of MYADM in cell lines representing target tissues of PeV and in intestinal organoids abolished PeV infection, demonstrating the importance of this factor in a physiologically relevant system. The authors performed a detailed analysis of interspecies sequence differences in MYADM orthologs, and identified the fourth extracellular loop of MYADM a likely site of virus attachment. They also showed that a single amino acid substitution in this region is responsible for the inability of a horse MYADM ortholog to mediate PeV infection. This is valuable information as it could help inform the design of genetically engineered animal models of PeV infection (e.g., using “humanized” animal models through modified receptor genes), which have not been available. Finally, the authors discovered that MYADM did not function in the cell surface attachment or internalization of PeV, but instead may function after endosomal acidification. The study is supported by strong phenotypes in physiologically relevant cell tissue culture models, as well as a detailed analysis of the role of MYADM in PeV species tropism. The paper is well written, its experiments are carried out well, and its findings are of broad interest.

Major:

1. Fig 2 and Extended Data Fig 5: Can the authors provide expression levels of MYADM in wild-type and MYADM KO organoids? E.g. Western blots, immunostaining.
2. Fig 3g and Extended Data Fig 8: Can the authors provide expression levels of MYADM ortholog mutants to show that mutants that did not rescue PeV infection in MYADM KO cells were indeed expressed?
3. Fig 4b,c and Extended Data Fig 10: PeV is unable to infect MEFs solely due to the lack of a compatible MYADM ortholog, but expression of human ITGB6 in MEFs led to virion internalization. Does this internalization lead to productive infection in the absence of MYADM, or is a compatible MYADM ortholog strictly required for productive infection?
4. Fig 4 and Extended Data Fig 11: MYADM only interacts with PeV capsid at slightly acidic pH, while initial cell surface attachment and internalization into the endosome are mediated by integrins. Combined with results on replicons, these findings would tentatively place MYADM's role at the virion uncoating/endosome escape step. However, the importance of MYADM-PeV interaction is still

relatively unclear. In Extended Data Fig 11, the images of MYADM KO cells incubated with virions at 37C for 6h appears to show a removal of internalized viral RNA compared to 30 min and 0 min. Could it be the case that without MYADM, PeV particles become stuck in the endosome and eventually get degraded in lysosomes? Somehow, finding a way to quantify the information shown microscopy images would have been useful. Otherwise, a more quantitative assay along the same vein as Extended Data Fig 11, such as RT-qPCR of intracellular viral RNA at various time points before and after when uncoating is expected to happen in wild-type cells, could better define the role of this protein in PeV entry (e.g., one would expect to see eventual degradation of viral RNA that initially accumulates in MYADM deficient cells after being brought in by particles).

5. Line 208 – “A recent study independently found MYADM as a host factor for multiple PeV-A strains.” For clarity, the authors should clarify which strains are overlapping with the strains being studied here in their manuscript. Also, could the authors note any differences between their findings and those of the prior study? It would be useful for the authors to comment on the aspects of their study that complement those of the prior study (e.g., specifically, on what new information is being reported here?).

Minor:

1. In some cases, letters in figures are all lowercase, but some in figure legends are uppercase.

3. Error bars are not visible in Fig 1b. Could the authors confirm error bars are shown here and are just smaller than the symbols?

4. Extended Data Fig 7: Could the authors annotate the alignment, showing where the key extracellular loop is located (and optionally the other helices and extracellular loops)?

Reviewer #2 (Remarks to the Author):

This manuscript reports on the identification of MYADM as a receptor involved in the entry of human par echoviruses (PeVs). Using a CRISPR-based screen in intestinal HT-29 cells, the authors identify MYADM in the top 10 highest scoring genes. The authors confirm these findings in a number of ways,

including generating primary intestinal organoids deficient in MYADM, defining the role of the extracellular loop in of MYADM in PeV infection, and performing biochemical assays to assess the possible interaction between MYADM and PeV particles under acidic and neutral conditions. Based on these data, the authors conclude that MYADM plays a role in PeV entry post-attachment, most likely at the stage following endocytosis. While it is clear from the data that the authors have identified MYADM as a host factor involved in PeV infections post-entry, their data do not fully support their conclusions that this factors is a receptor or engages particles in a pH-dependent manner. My specific concerns and/or suggestions are found below:

General comments:

1. The use of HT-29 cells lacking key surface receptors is less than ideal. While the authors provide a justification for this, one wonders what the impacts of this could have on their overall screening specificity. This is further exemplified by "hits" from the screen that would not even be expressed in the cell types used (e.g., CGB8).
2. MYADM is known to maintain junctional integrity in cells, often via actin signaling. It is important to demonstrate that this does not explain the phenotype the authors observe, which would be independent of viral entry directly. What is the junctional integrity in MYADM KO cells, as assessed by immunostaining and functional assays (e.g., TER measurements). Perhaps MYADM deletion is simply exposing an otherwise non-accessible receptor, which would explain many of the observed phenotypes.
3. Much of the study relies on reporter virus readouts as surrogates for bona fide infection assays. There are many issues with this approach, including the very high background of many of these assays. For example, in Figure 1b, the background luciferase activity is 10^6 , which makes it very difficult to determine whether there may be low levels of replication occurring below this background. These data should be supplemented with assays for viral infectivity, such as plaque assays and/or TCID50 assays.
4. In several figures, the scale of the y-axis is set at "Relative RNA", often normalized to some control (e.g., Figure 1c, 1d, 2b, 4e). This also limits an analysis of how much active replication is occurring in cells deficient in MYADM. These data should be supplemented with bona fide readout of viral infectivity.
5. Additional controls in Figure 1 and 2 would better show the specificity of the effects of MYADM deletion on pan-picornavirus replication. Data showing the levels of infection of related viruses whose proteinaceous receptors are known (e.g., CV-B, PV) would bolster these findings.
6. Immunoblots from KO cells lines would better support the knockout in these cells.
7. The data in Figure 4 do not support the direct interaction between MYADM and PeV, not are they sufficient to support the role of pH in this process. The authors first state that KO of MYADM had no effect on binding or internalization. Thus, MYADM is not an entry receptor. They then go on to apply

IP-based approaches to suggest an interaction using RT-qPCR at two time point post-infection. These data do not support an interaction between PeV particles and MYADM. Moreover, there are no additional data to support any possible interaction (e.g., imaging-based approaches, in vitro biochemical assays).

8. The data in Figure 4 also do not support a pH dependence for MYADM binding, given that these studies are all performed in vitro and the stability of PeV particles is not shown. Are PeV particle incubated at pH of 5.5. stable? If not, the assay is not suitable to support a pH dependence. Additional data could be obtained using imaging of viral internalization or other more direct assays (e.g, inhibition of acidification in cells).

Reviewer #3 (Remarks to the Author):

Qiao et al reported a decent study of identifying MYADM to be a cellular receptor of human parechoviruses. In CRISPR/Cas genetic screens in PeV-A1 of PeV-A2 infected cells, MYADM and ITGB6 were among the top hits. The authors then performed a series of experiments in cell lines and human colon organoids and demonstrated the essential role of MYADM for virus infection and viral entry. They also identified the critical amino acid residues in MYDAM mediating its binding with PeV-A1. The study was well designed with lots of data. The findings of this study are novel and original. Statistical analysis was properly conducted. Most presented data are quite convincing.

However, several issues should be addressed or improved.

The authors used an excellent, biological-relevant in vitro model, human colon organoids, in combination with genetic knockout in these organoids, to demonstrate the essential role of MYADM for PeV-A1 infection. I believe the data would be ever stronger if the authors use colon organoids in more experiments and highlight the application of organoids. E.g. PeV-A3 and/or A2 infection in the wild and MYADM knock-out organoids.

The demonstration of viral infection in colon organoids is a very strong evidence for the conclusion, given the biological relevance of epithelial organoids and human enteric tropism of parechoviruses. Apart from the infection data shown in Fig. 2 and the aforementioned experiments, I suggest the authors use a viral antigen-specific antibody to label virus-infected cells, rather than an anti-dsRNA. IF staining using a viral antigen-specific antibody is a more convincing data and usually the image quality will be greatly improved. In addition, the authors may show the distribution/abundance of MYADM in wildtype and knockout organoids. In Fig. 2a, not all dsRNA signals are within the cytosol, which is not reasonable.

MYADM binding to PeV-A1 particles was dependent on slightly acidic conditions representative of early and late endosomes as shown by cell-free immunoprecipitation assay. It would be more convincing if the authors could show data using endosomal acidification inhibitors (such as chloroquine, bafilomycin A1 and NH₄Cl) in cell lines, and ideally organoids.

Minor issues

1. Line 101. The authors used apical-out intestinal organoids for infection experiments and cited the paper. However, they may not describe the preparation of these apical-out organoids in Methodology.
2. Line 99. "cell models" is not a good description for organoids. consider using "biologically relevant in vitro models".

REVIEWER COMMENTS

We thank the reviewers of our manuscript for the insightful suggestions and overall positive evaluations. Based on these thoughtful suggestions from the reviewers we have addressed their concerns and included a substantial amount of new data to enhance the strength and robustness of our study. In particular, we provide more data supporting the role of MYADM as receptor for PeV by showing that PeV-A1 particles are highly stable under the conditions tested, thereby addressing the main criticism on the in vitro binding assay mentioned by one of the reviewers. In addition, we now included additional controls showing that the binding in the in vitro assays as well as binding in the cellular assay, is specific for PeV-A1 because we observed no interaction with the tested enterovirus (EV-A71). We already showed that specific mutations in the extracellular loop of MYADM that disrupt binding in our assays also prevent functional rescue of PeV-A infection when expressed in MYADM knockout cells. The improved binding data, combined with the replicon data that shows that the strong phenotype of MYADM knockout cells is specific to the viral entry step, points to a role of MYADM as essential receptor. The observation that cellular attachment is mediated by integrins and not MYADM does not mean that MYADM is not a receptor. For picornaviruses it is not uncommon that the function of primary cell attachment is mediated by a separate attachment receptor, while post attachment steps are mediated by the cognate receptor (PMID: 36307169). In addition to including more data that increase the rigor of the binding assays, we have included more virological assays (plaque assays) and improved the reporter virus assays to characterize the effect of MYADM knockout. These sensitive and quantitative assays confirmed the very strong phenotype of MYADM knockout highlighting the essential role of MYADM in virus infection. Finally, we have now also included another PeV-A genotype in the organoid cultures and added various controls requested by the reviewers. We thank the reviewers for raising very relevant points and we believe that by experimentally addressing these points, we have significantly improved our manuscript.

The revised manuscript has been expanded and re-organized into 4 main figures, 18 extended data figures, and 1 supplementary table. Throughout the rebuttal we mention the corresponding figure panel of the revised version of the manuscript when applicable. Please find below the reviewers' comments in full with our responses in italic (blue).

Reviewer #1 (Remarks to the Author):

Qiao et al. used a genome-wide CRISPR-Cas9 screen to identify MYADM as an essential host factor involved in paraechovirus (PeV-A) entry. Genetic depletion of MYADM in cell lines representing target tissues of PeV and in intestinal organoids abolished PeV infection, demonstrating the importance of this factor in a physiologically relevant system. The authors performed a detailed analysis of interspecies sequence differences in MYADM orthologs, and identified the fourth extracellular loop of MYADM a likely site of virus attachment. They also showed that a single amino acid substitution in this region is responsible for the inability of a horse MYADM ortholog to mediate PeV infection. This is valuable information as it could help inform the design of genetically engineered animal models of PeV infection (e.g., using “humanized” animal models through modified receptor genes), which have not been available. Finally, the authors

discovered that MYADM did not function in the cell surface attachment or internalization of PeV, but instead may function after endosomal acidification. The study is supported by strong phenotypes in physiologically relevant cell tissue culture models, as well as a detailed analysis of the role of MYADM in PeV species tropism. The paper is well written, its experiments are carried out well, and its findings are of broad interest.

Major:

1. Fig 2 and Extended Data Fig 5: Can the authors provide expression levels of MYADM in wild-type and MYADM KO organoids? E.g. Western blots, immunostaining.

We have performed the Western blot suggested by the reviewer, confirming the knockout of MYADM in MYADM KO organoids. The result was added to Extended Data Figure 7b. Additionally, we attempted immunostaining using the same MYADM antibody we used for detecting MYADM knockout in cell lines via western blot. However, the antibody resulted in nonspecific staining in both cell lines and organoids. Subsequently, we tested three commercially available antibodies for MYADM (Abcam ab212997; Bioss, BS-19143R; and Mybiosource, MBS669098) and none of them proved effective for immunostaining. Giving the combined evidence from our genotyping and the newly added Western blot results, we are confident in confirming the successful knockout of MYADM in the organoids.

Extended Data Figure 7. Validation of human colonoids MYADM isogenic knockout. b, Western blot analysis of MYADM expression in organoids.

2. Fig 3g and Extended Data Fig 8: Can the authors provide expression levels of MYADM ortholog mutants to show that mutants that did not rescue PeV infection in MYADM KO cells were indeed expressed?

We have performed the western blot suggested by the reviewer, showing all the MYADM ortholog mutants are indeed expressed in MYADM KO cells. The results are now shown in Extended Data Figure 11c.

Extended Data Figure 11. c, Western blot analysis demonstrating the expression of the corresponding HA-tagged MYADM variants in HT29-DKO Δ MYADM cells. GAPDH was used as a loading control.

3. Fig 4b,c and Extended Data Fig 10: PeV is unable to infect MEFs solely due to the lack of a compatible MYADM ortholog, but expression of human ITGB6 in MEFs led to virion internalization. Does this internalization lead to productive infection in the absence of MYADM, or is a compatible MYADM ortholog strictly required for productive infection?

We thank the reviewer for highlighting this point. In Fig 3a and Extended Data Fig 9, we demonstrated that MEF and BHK21 cells become susceptible to PeV infection when human MYADM is expressed. In contrast, expression of human ITGB6 in MEF and BHK21 cells does not lead to productive infection, despite enhancing binding and internalization of PeV. This suggests a strict requirement of a compatible MYADM ortholog for PeV infection.

4. Fig 4 and Extended Data Fig 11: MYADM only interacts with PeV capsid at slightly acidic pH, while initial cell surface attachment and internalization into the endosome are mediated by integrins. Combined with results on replicons, these findings would tentatively place MYADM's role at the virion uncoating/endosome escape step. However, the importance of MYADM-PeV interaction is still relatively unclear. In Extended Data Fig 11, the images of MYADM KO cells incubated with virions at 37C for 6h appears to show a removal of internalized viral RNA compared to 30 min and 0 min. Could it be the case that without MYADM, PeV particles become stuck in the endosome and eventually get degraded in lysosomes? Somehow, finding a way to quantify the information shown microscopy images would have been useful. Otherwise, a more quantitative assay along the same vein as Extended Data Fig 11, such as RT-qPCR of intracellular viral RNA at various time points before and after when uncoating is expected to happen in wild-type cells, could better define the role of this protein in PeV entry (e.g., one would expect to see eventual degradation of viral RNA that initially accumulates in MYADM deficient cells after being brought in by particles).

We agree with the reviewer that our findings suggest MYADM may play a role in the virion uncoating/endosome escape step. In Fig 4, using replicon and binding/internalization assays, we demonstrated the critical role of MYADM in PeV-A1 post-attachment steps. The FISH assay in Extended Data Fig 11 (updated as Extended Data Fig 14) provided supplementary support for the binding/internalization assays, showing that PeV-A1 could enter cells even in the absence of

MYADM. Our interpretation of the FISH assay, which is based on our experience with the timeline of virus infection, is that the strong signal observed in the wild-type cells at 6 hours post infection corresponds to new RNA produced by viral RNA replication rather than incoming viral RNA. Quantitation of the FISH results is technically challenging and the high MOIs required for these imaging experiments may complicate interpretation. We appreciate the reviewer's advice on tracking intracellular viral RNA levels by RT-qPCR at various time points. Following the reviewer's suggestions, we have quantified viral RNA levels at multiple time points in WT and MYADM KO HT29-DKO cells by RT-qPCR. We found that the RNA levels in wild-type cells increased starting at 2.5 hours post infection, marking the beginning of RNA replication of the incoming viral RNA. We observed no differences in viral RNA levels between wild-type and MYADM knockout cells at time points earlier than 2.5 h.p.i. This is consistent with our previous binding/internalization assay performed in A549 cells, where MYADM exhibited no observable impact on virus binding and internalization at early time points. in HT29-DKO cells

Quantification of viral RNA levels using binding and internalization assays by RT-qPCR at various time points in HT29-DKO WT and MYADM KO cells.

It should be noted that the FISH assay nor the RT-qPCR assay directly assess virion uncoating/endosome escape. It is unclear if preventing virion uncoating/endosome escape would lead to rapid degradation of the incoming virion. There are several fundamental difficulties associated with characterizing uncoating/endosome escape using imaging-based assays: (1) the high particle-to-plaque forming unit (pfu) ratio of picornaviruses (e.g. 100:1 to 200:1 for poliovirus) makes it difficult to distinguish productive from nonproductive pathways, (2) visualization assays generally require a very high multiplicity of infection (MOI), increasing the probability of sampling nonproductive pathway, (3) capturing when the viral RNA enters the cytosol and distinguishing this event from still encapsidated RNA is technically challenging. Therefore, in manuscripts dedicated to demonstrating that a receptor is an uncoating receptor for picornaviruses the approaches are almost exclusively based on structural biology approaches such as Cryo-EM and X-ray crystallography (e.g. PMID: 30478256, PMID: 31104841, PMID: 26563423). These would be very valuable future endeavors, but we believe that they are outside of the scope of the current manuscript. While we have not shown (nor claim to have shown) that MYADM is a receptor specifically involved in uncoating, we have shown that MYADM interacts with PeV-A1 particles in a cellular assay and an in vitro assay, and we have shown that specific mutations in the extracellular loop of MYADM strongly reduce both binding as well as infection. This demonstrates the importance of the MYADM-PeV interaction. Combined with the replicon data and the strong

phenotype specific to viral entry, we provide multiple lines of evidence that point to a role of MYADM as an essential receptor for parechovirus infection.

5. Line 208 – “A recent study independently found MYADM as a host factor for multiple PeV-A strains.” For clarity, the authors should clarify which strains are overlapping with the strains being studied here in their manuscript. Also, could the authors note any differences between their findings and those of the prior study? It would be useful for the authors to comment on the aspects of their study that complement those of the prior study (e.g., specifically, on what new information is being reported here?).

We have expanded on the description of the main findings from another study, addressing the reviewer’s suggestion: “A recent study independently found MYADM as a host factor for PeV-A3 as well as five other PeV-A genotypes (PeV-A1, 2, 4-6) and showed the interaction between MYADM and the VP0 capsid protein of PeV-A3²⁷.” Our discussion section emphasizes the novel findings in our study compared to the prior one: 1) we establish the relevance between MYADM and PeV-A infection in diverse cell types including a physiologically relevant organoid model system; 2) our mutagenesis mapping studies pinpoint specific residues within the fourth extracellular loop critical for receptor function and binding; 3) we demonstrate that MYADM acts as a cellular receptor by demonstrating binding to viral particles.

Minor:

1. In some cases, letters in figures are all lowercase, but some in figure legends are uppercase.

We have corrected the letters in figure legends and indicated the correction in the manuscript.

3. Error bars are not visible in Fig 1b. Could the authors confirm error bars are shown here and are just smaller than the symbols?

The error bars in Fig 1b are indeed smaller than the symbols. We have updated Fig. 1b using low background PeV-A1-nluc and reduced the symbol size to ensure the visibility of the error bars. Additionally, we addressed sample numbers (n = 3) in the figure legend.

4. Extended Data Fig 7: Could the authors annotate the alignment, showing where the key extracellular loop is located (and optionally the other helices and extracellular loops)?

We have updated Extended Data Fig 7 and figure legend (updated as Extended Data Fig 10) following the reviewer's suggestions. The key extracellular loop is highlighted by a solid box and the predicted helices are shown above the sequences.

Extended Data Figure 10. Sequence alignment of full-length MYADM proteins from various mammalian species. The key extracellular loop is highlighted by a solid box and the predicted helices are shown above the sequences.

Reviewer #2 (Remarks to the Author):

This manuscript reports on the identification of MYADM as a receptor involved in the entry of human par echoviruses (PeVs). Using a CRISPR-based screen in intestinal HT-29 cells, the authors identify MYADM in the top 10 highest scoring genes. The authors confirm these findings in a number of ways, including generating primary intestinal organoids deficient in MYADM, defining the role of the extracellular loop in of MYADM in PeV infection, and performing biochemical assays to assess the possible interaction between MYADM and PeV particles under acidic and neutral conditions. Based on these data, the authors conclude that MYADM plays a role in PeV entry post-attachment, most likely at the stage following endocytosis. While it is clear from the data that the authors have identified MYADM as a host factor involved in PeV infections post-entry, their data do not fully support their conclusions that this factors is a receptor or engages particles in a pH-dependent manner. My specific concerns and/or suggestions are found below:

General comments:

1. The use of HT-29 cells lacking key surface receptors is less than ideal. While the authors provide a justification for this, one wonders what the impacts of this could have on their overall

screening specificity. This is further exemplified by "hits" from the screen that would not even be expressed in the cell types used (e.g., CGB8).

We have used HT-29 cells because they are derived from primary colorectal adenocarcinoma, and parechoviruses are known to infect the intestinal tract. We chose to perform the screens in HT29 cells lacking heparan sulphate and sialic acid (named HT29 double knockout, HT29-DKO) to favor identification of cognate protein receptors, as viruses sometimes adapt in tissue culture to utilize heparan sulphate and/or sialic acid (PMID: 31266258). The most strongly enriched genes were MYADM and ITGB6 in screens using PeV-A1 and PeV-A2. The identification of the known attachment factor ITGB6 provides confidence in the screening specificity. We have rigorously validated the role of MYADM in many different cell lines and organoid cultures further providing confidence that the screening strategy in HT29-DKO cells was valid. We have not followed up on the genes that scored lower in significance and fold-enrichment than MYADM and ITGB6 and we cannot exclude that they contain false positives (as in any genetic screen). Whether screens in HT29 cell expressing heparan sulphate and sialic acid would have generated more (or less) false positives is unknown. However, given the extensive validation of MYADM in a plethora of different cell types including primary cells, we believe that the screens in HT29-DKO cells have been a sound discovery tool.

2. MYADM is known to maintain junctional integrity in cells, often via actin signaling. It is important to demonstrate that this does not explain the phenotype the authors observe, which would be independent of viral entry directly. What is the junctional integrity in MYADM KO cells, as assessed by immunostaining and functional assays (e.g., TER measurements). Perhaps MYADM deletion is simply exposing an otherwise non-accessible receptor, which would explain many of the observed phenotypes.

The reviewer brings up an interesting point raising the possibility that MYADM affects PeV-A infection by its role in maintaining junctional integrity in cells. Based on the below reasons this is a highly unlikely scenario:

1) If MYADM would act by maintaining junctional integrity, one would expect that the phenotype would only be observed in cell systems where endothelial/epithelial junctions would in fact be formed. This includes 2D cultures of polarized epithelium/endothelium and 3D organoid cultures. While we do observe that MYADM knockout affects infection of intestinal organoids, which are known to form junctions, we observe an equally strong phenotype in multiple cancer cell lines grown under standard culture conditions. Because it is well established that these cell lines do not form tight junctional barriers (PMID: 24079544; PMID: 3391241), it is highly unlikely that the strong phenotype of MYADM knockout on infection is due to affecting barrier function.

*2) The reviewer poses that "perhaps MYADM deletion is simply exposing an otherwise non-accessible receptor, which would explain many of the observed phenotypes". We disagree with this statement because if this was the case, we would expect the opposite phenotype. We would expect that MYADM knockout in cell that form tight junctions (such as the intestinal organoids) would lead to an **increased** susceptibility. Our results show that knockout of MYADM results in a loss of infectivity.*

The strong effect of MYADM knockout in multiple cells systems, its role in mediating functional entry, and the observed specific binding of MYADM with PeV-A particles are all consistent with a direct role of MYADM as essential receptor.

3. Much of the study relies on reporter virus readouts as surrogates for bona fide infection assays. There are many issues with this approach, including the very high background of many of these assays. For example, in Figure 1b, the background luciferase activity is 10^6 , which makes it very difficult to determine whether there may be low levels of replication occurring below this background. These data should be supplemented with assays for viral infectivity, such as plaque assays and/or TCID50 assays.

We thank the reviewer for pointing out the potential issue with background in the luciferase reporter assays and appreciate the comment to use additional infectivity assays such as plaque assays. We have taken steps to address these two concerns:

1) During our work with using the reporter virus, we noticed that the residual nanoluciferase from the producers cells could lead to background when preparing the viral stock. This was variable depending on the specific viral preparation and the time of harvest. For example, the viral preparation used in Fig. 3g resulted in a very low background (10^3) while (as the reviewer rightfully noticed) the preparation used for Fig 1b. had a high background (10^6). We agree with the reviewer that this makes it more difficult to determine whether low levels of replication could still occur. Therefore, we have now adjusted our protocol in viral stock preparation by removing background nano-luciferase using ultracentrifugation. The pelleted virus particles were resuspended in clean culture media. This adjustment has resulted in a significant reduction in background signals (10^2). This signal is comparable to the signal observed when we use the nanoluciferase reagents on cells that are not infected. We have repeated the experiment in Figure 1b with this viral stock and updated this in the revised manuscript. The low background now allows us to exclude that MYADM knockout cells support low levels of viral replication and highlights the strong dependence on MYADM for PeVA infection (10^5 reduction in MYADM knockout cells).

b

Fig.1 MYADM is an essential host factor for human parechoviruses. b, Time course of nLuc-expressing PeV-A1 infection in HT29-DKO and HuTu80 wild type (WT), MYADM knockout (Δ MYADM), and HA-tagged MYADM-complemented MYADM knockout (Δ MYADM + MYADM-HA) cells at MOI 0.1 (mean \pm SEM, n = 3).

2) We have performed the additional virological assays requested by the reviewer. This data is now included in the revised manuscript as a new extended data figure 5. Reassuringly, the plaque assays and the viral reporter assays are highly congruent with each other (e.g. compare the time course in HT29-DKO and HuTu80 in Fig. 1b with new extended data figure 5a and the time course in HEK293FT in Fig 4b with the new data figure 5a).

Extended Data Figure 5. Quantification of PeV-A1 and PeV-A3 infection in various MYADM or ITGB6 cell lines across multiple cell types. A, HT29-DKO, HuTu80, HEK293FT, and A549 cell lines were infected with PeV-A1 at MOI 0.1, virus titers were quantified by plaque assay at the indicated time points (mean \pm SEM, n = 3). B, HuTu80, HEK293FT, and A549 cell lines were infected with PeV-A3 at MOI 0.1, virus titers were quantified by plaque assay at the indicated time points (mean \pm SEM, n > 3).

4. In several figures, the scale of the y-axis is set at “Relative RNA”, often normalized to some control (e.g., Figure 1c, 1d, 2b, 4e). This also limits an analysis of how much active replication is occurring in cells deficient in MYADM. These data should be supplemented with bona fide readout of viral infectivity.

Throughout the manuscript in our original submission, we have used several methods to analyze how much active replication is occurring in cells deficient in MYADM. One of the methods is

reverse transcription quantitative PCR (RT-qPCR). This allows us to quantify the level of positive stranded viral RNA in the cell. During infection, viral RNA levels increase indicative of active viral RNA replication and RT-qPCR measures this increase (see e.g. Fig 2b). Inherent to the RT-qPCR technique the measurement is a relative measurement and needs to be normalized. We have indicated in the figures to which is normalized. Although we believe that RT-qPCR for viral RNA levels is a bona fide readout for viral RNA replication, we agree with the reviewer that additional virological assays to complement these results are valuable. We have now repeated several experiments looking at the effect of MYADM knockout on viral RNA replication and particle production using plaque assays. We have included this in the new extended data figure 5. The plaque assays further highlighted that MYADM is essential for viral infection. Reassuringly, the experiments with plaque assays are highly congruent with the RT-qPCR (e.g. compare fig. 1c, and 1d with new extended data figure 5a, b, and c).

Throughout the manuscript we have used complementary approaches to analyze how much replication is occurring in wild type cells, MYADM knockout and addback cells. We have used reporter viruses, RT-qPCR for viral RNA replication, immunostainings for viral antigens, crystal violet staining for virus induced c.p.e., and plaque assays. Thanks to the comments of the reviewer, we have now strongly reduced the background of the reporter virus in Fig. 1b and have markedly expanded the use of plaque assays. Together the new experiments have further improved the rigor of the manuscript and highlighted the strong dependence of PeVA on MYADM through sensitive and quantitative assays.

5. Additional controls in Figure 1 and 2 would better show the specificity of the effects of MYADM deletion on pan-picornavirus replication. Data showing the levels of infection of related viruses whose proteinaceous receptors are known (e.g., CV-B, PV) would bolster these findings.

In response to the reviewer's comment, we conducted additional experiments to address the specificity of MYADM deletion on pan-picornavirus replication. We tested EV-A71 and CV-B3 reporter viruses in WT and MYADM KO HEK293FT cells at MOI 0.1 and harvested the cells at a timepoint when efficient replication had occurred (24 h.p.i.). In stark contrast to experiments with human parechoviruses, no significant decreases in virus replication were observed in MYADM KO cells (new Extended Data Figure 6). Additionally, in previous loss-of-function genetic screens for several picornaviruses other than PeV-A, MYADM is not identified as a proviral host factor (PMID: 31527793, PMID: 29681460, PMID: 28077878, PMID: 31409686). Together, this supports the specificity of the effects of MYADM deletion, further highlighting its critical role for human parechoviruses.

Extended Data Figure 6. MYADM dependency of different picornaviruses. WT and MYADM KO HEK293FT cells were infected with EV-A71 and CV-B3 reporter viruses at MOI 0.1 for 24 hours (mean ± SEM, n > 4, paired t test; ns, not significant).

6. Immunoblots from KO cells lines would better support the knockout in these cells.

To validate the gene knockout, we performed genotyping for all the KO cell lines and organoids in Extended Data Figure 2 and 7. In addition, western blot analysis of MYADM knockout and addback cell lines was provided in Extended Data Figure 3. Furthermore, we have now incorporated a western blot to confirm MYADM knockout in human organoids in Extended Data Figure 7b. In our attempt to perform western blot analysis of ITGB6 knockout cells, the antibody failed to detect endogenous ITGB6 expression. However, the cDNA addback of ITGB6 successfully reversed the induced phenotype of ITGB6 KO, providing further confirmation of the DNA genotyping results, which showed the successful generation of a frameshift mutation in the ITGB6 gene.

Extended Data Figure 7. Validation of human colonoids MYADM isogenic knockout. b, Western blot analysis of MYADM expression in organoids.

7. The data in Figure 4 do not support the direct interaction between MYADM and PeV, not are they sufficient to support the role of pH in this process. The authors first state that KO of MYADM had no effect on binding or internalization. Thus, MYADM is not an entry receptor. They then go on to apply IP-based approaches to suggest an interaction using RT-qPCR at two time point post-infection. These data do not support an interaction between PeV particles and MYADM. Moreover, there are no additional data to support any possible interaction (e.g., imaging-based approaches, in vitro biochemical assays).

We appreciate the reviewer's valuable feedback and would like to address the concerns regarding the interpretation of the data in Figure 4 and the proposed interaction between MYADM and PeV.

The reviewer states that “They then go on to apply IP-based approaches to suggest an interaction using RT-qPCR at two time point post-infection. These data do not support an interaction between PeV particles and MYADM.” It is unclear to us what the reviewer means by this statement. Co-immunoprecipitation is a widely used technique to demonstrate interactions. We chose to use very early time points during infection to capture interactions between MYADM and incoming PeV-A1 particles. We observed a clear co-immunoprecipitation of PeV-A1 particles (as measured using RT-qPCR for viral RNA), Co-immunoprecipitation with a mutant of MYADM with three mutations in MYADM’s extracellular resulted in a strong reduction of this co-immunoprecipitation. Given the reviewer’s prior comments on the specificity of MYADM for PeV-A1, we have now added additional data in this assay using EV-A71 infection. For EV-A71 (Extended data figure 16), We did not observe co-immunoprecipitation of EV-A71 with MYADM. We believe that our data supports an interaction between PeV-A1 particles and MYADM and the new data now highlights the specificity of the interaction.

The reviewer states: “Moreover, there are no additional data to support any possible interaction (e.g., imaging-based approaches, in vitro biochemical assays).” We are unclear what the reviewer means by this statement because we have used in vitro pull-down assays to provide additional data to support the interaction. Perhaps this statement is based on technical concerns raised by the reviewer in the next comment on the in vitro assay and particularly the concern that the acidic pH might affect particle stability. We have now provided additional data to address these concerns and validate the reliability of our in vitro results accordingly (see next comment).

Extended Data Figure 16. Time point virus particle binding assay. Vero cells, Vero cells expressing HA-tagged human MYADM (Vero + WT), and 3AA-EHN mutant (Vero + 3AA-EHN) were incubated with concentrated EV-A71 at 4 °C for 1 h, followed by incubation at 37 °C for 0, 15, or 30 min after removing unbound viruses. Cells were lysed (Input) and immunoprecipitated with anti-HA beads (IP). Viral RNA levels were measured by RT-qPCR (Mean ± SEM, n > 3, two-way ANOVA with Šídák’s multiple comparison test; ns, not significant).

8. The data in Figure 4 also do not support a pH dependence for MYADM binding, given that these studies are all performed in vitro and the stability of PeV particles is not shown. Are PeV particles incubated at pH of 5.5. stable? If not, the assay is not suitable to support a pH

dependence. Additional data could be obtained using imaging of viral internalization or other more direct assays (e.g, inhibition of acidification in cells).

Following the reviewer's last comment, our in vitro binding assay serves as additional evidence supporting the interaction between MYADM and PeV. While no pull down was observed in slightly alkaline conditions (pH-7.4), we observed readily detectable pull down at the (slightly) acidic conditions of pH-6.5, 6.0 and 5.5. The reviewer's concern is that if the particle is not stable, the observed interaction might be aspecific. To address this, we conducted additional validation of virus stability after incubation in our specific IP buffer (NETN buffer) with varied pH under the same condition for the IP experiment. Our plaque assay results showed no difference in virus particle stability across different pH in the IP buffer. This is fully consistent with the known literature that report that parechoviruses, like many enteroviruses and unlike rhinoviruses, are highly stable at low pH (PMID: 17553229) being refractory to acid inactivation even at a pH much lower (pH-3, PMID:17553229) than used in our assays.

Virus particle stability of gradient PeV-A1-Flag virus across different pH in the NETN buffer used for the in vitro virus binding assay.

The stability of the particles at the pH range at which we performed the in vitro pull-down assays provides further confidence in the results. In our original submission, we already included the MYADM-3AA-EHN mutant as control that the pull down was not due to promiscuous binding of PeV-A1 particles to the beads. To exclude that MYADM is binding aspecifically to other viruses, we repeated the pulldown using EV-A71 particles. We observed no binding between MYADM and EV-A71 and this control is now also included in the revised manuscript (Extended Data Figure 18). We believe that this data further confirms the validity and specificity of the in vitro binding assay. Similar binding assays relying on pull down have been used in the established literature for other picornaviruses to provide evidence that the identified entry factor is receptor (e.g. PMID: 30808762, PMID: 29681460). The observation that cellular attachment is mediated by integrins and not MYADM does not mean that MYADM is not a receptor. For picornaviruses it is not uncommon that the function of primary cell attachment is mediated by a separate attachment receptor, while post attachment steps are mediated by the cognate receptor (PMID: 36307169).

Extended Data Figure 18. The pH-dependent virus binding assay of MYADM with PeV-A1-Flag (a) and EV-A71 (b). Left, western blot analysis of the input viruses in NETN buffer, as well as the final immunoprecipitates (IP) of HA-tagged MYADM proteins and viral proteins. Right, quantification of viral RNA levels after immunoprecipitation using RT-qPCR (Mean \pm SEM, $n > 3$).

We agree that the pH dependence for MYADM binding is mainly based on the *in vitro* experiments. We have also observed binding in a cell entry assay IP experiment at early time-points. Although the time points correspond to uptake into the endocytic system, which is known to lead to acidification, we agree with the reviewer that additional experiments would be helpful. A similar point was made by reviewer 3. Following the reviewers' recommendation, we conducted time course immunoprecipitation assays (IP) with endosomal acidification inhibitors, bafilomycin A1 (BafA1) and NH₄CL, in Vero cells. The same IP assay was performed with samples after 15 min internalization, but pretreated with 100 nM BafA1 or 30 mM NH₄CL for 1 hour at 37°C and along the whole process. While the results confirmed the binding of PeV-A1 protein with MYADM during entry, we observed no significant difference in inhibitor-treated cells (shown below). Although this could mean that endocytic acidification in cells is not required for MYADM's interaction with PEV-A1, the observation that *in vitro* the interaction already occurs at pH-6.5 and is optimal at pH-6.0 leaves open the possibility that BafA1 or NH₄CL treatment did not prevent the slight acidification typical for early endosomes. For example, in a previous study with A431 cells, the intralysosomal pH increased from about 5.1-5.5 to about 6.3 in the presence of BafA1 (PMID: 1832676). The lack of tools to completely prevent acidification (and the likely cell toxicity

that would be associated with that) limits our ability to perform these experiments. To avoid the impression that we have conclusively proven the pH dependence in cells we have removed the indication of the endosomal pH in the schematic model in fig. 4 and we have clarified in the discussion what we can conclude from the current data: “We demonstrated that MYADM binds to PeV-A1 particles and is essential for functional viral entry, which points to a role as receptor. MYADM knockout did not reduce cell binding, suggesting that it does not act as a primary cellular attachment factor for PeV-A1. Rather, binding of PeV-A1 to MYADM during viral entry was enhanced on raising the temperature to allow endocytosis. Moreover, the *in vitro* binding between MYADM with parechovirus particles was detected at slightly acidic conditions (pH ≤ 6.5). Whether during a viral infection the binding is similarly dependent on the pH will need to be experimentally validated because *in vivo* and *in vitro* binding conditions can differ.”

Time point virus particle binding assay. Vero cells and Vero cells expressing HA-tagged human MYADM (Vero + WT) were pretreated with 100 nM BafA1 or 30 mM NH₄CL for 1 hour at 37 °C, incubated with concentrated PeV-A1 at 4 °C for 1 h, followed by incubation at 37 °C for 15 min after removing unbound viruses along with the compounds. Cells were lysed (Input) and immunoprecipitated with anti-HA beads (IP). Viral RNA levels were measured by RT-qPCR.

Besides the specifics of the pH-dependence in cells, we want to highlight that our replicon, binding/internalization assays, and FISH assay collectively demonstrate the crucial role of MYADM in the post-attachment entry steps. Moreover, our time-course and IP assays, along with an *in vitro* PeV-MYADM interaction assay, indeed reveal a specific interaction between MYADM and PeV-A1. The lines of evidence that MYADM is a receptor essential for human parechoviruses presented in our manuscript is equal to or surpasses other manuscripts describing the identification of picornavirus receptors that similarly rely on pull down experiments to shown interaction of the receptor with viral particles (e.g. PMID: 30808762, PMID: 29681460). For picornaviruses it is quite common that the cognate receptor is not the receptor that mediates primary attachment to the cells (PMID: 36307169) and our data points to a role of MYADM as post attachment receptor. We thank the reviewer again for their comments that led to the incorporation of a significant amount of new data increasing the robustness and clarity of the manuscript.

Reviewer #3 (Remarks to the Author):

Qiao et al reported a decent study of identifying MYADM to be a cellular receptor of human parechoviruses. In CRISPR/Cas genetic screens in PeV-A1 of PeV-A2 infected cells, MYADM and ITGB6 were among the top hits. The authors then performed a series of experiments in cell lines and human colon organoids and demonstrated the essential role of MYADM for virus infection and viral entry. They also identified the critical amino acid residues in MYADM mediating its binding with PeV-A1. The study was well designed with lots of data. The findings of this study are novel and original. Statistical analysis was properly conducted. Most presented data are quite convincing.

However, several issues should be addressed or improved.

The authors used an excellent, biological-relevant in vitro model, human colon organoids, in combination with genetic knockout in these organoids, to demonstrate the essential role of MYADM for PeV-A1 infection. I believe the data would be ever stronger if the authors use colon organoids in more experiments and highlight the application of organoids. E.g. PeV-A3 and/or A2 infection in the wild and MYADM knock-out organoids.

We appreciate the reviewer's suggestion to expand the application of the organoid system to enhance the impact of our study. We have conducted additional experiments to further explore the role of MYADM in PeV-A2 and A3 infection using human colon organoids. In our extended experiments, we performed a time course infection of PeV-A2 and A3 in apical-out colonoids and quantified the infection using plaque assays. Compared to PeV-A1, PeV-A2 exhibited slower kinetics in WT colonoids, detectable at 96 hpi, while no infection was observed in MYADM KO (#22) colonoids (Extended Data Figure 8). For PeV-A3, efficient infection was not detected in WT colonoids, a result consistent with a previous study indicating that PeV-A3 infection in a primary intestinal epithelium 2D model could only be established with clinical isolates (cultured from stool specimens for one passage on a specific cell line) but not with laboratory-adapted PeV-A3 strains (PMID: 34790587).

Recognizing the importance of thoroughly investigating the role of MYADM in human colon organoids, we extended our study to include basal-out colonoids. Basal-out colonoids demonstrated a significantly enhanced infection with both PeV-A1 and A2 compared to apical-out colonoids (Extended Data Figure 8), in line with previous findings that PeV-A1 replication was highest with infection established from the basolateral side of a primary intestinal epithelium 2D model (PMID: 34790587).

In light of these new data, we have updated the manuscript to highlight the organoid orientation-dependent infection patterns of PeV-A1 and A2. Additionally, we have emphasized the relevance of these findings in the context of MYADM knockout, thereby strengthening the overall argument for the pivotal role of MYADM in PeV-A infection. We believe that these expanded experiments have addressed the reviewer's suggestion and also provide a more comprehensive understanding of PeV-A1 and A2 infection dynamics in the context of colon organoids.

Extended Data Figure 8. The infection of PeV-A1 and PeV-A2 in primary intestinal organoids. a, quantification of PeV-A2 infection in apical-out colonoids using plaque assay over time (mean ± SEM, n = 3). b, quantification of PeV-A1 and PeV-A2 infection in basal-out colonoids using RT-qPCR and plaque assay over time (mean ± SEM, n = 3).

The demonstration of viral infection in colon organoids is a very strong evidence for the conclusion, given the biological relevance of epithelial organoids and human enteric tropism of parechoviruses. Apart from the infection data shown in Fig. 2 and the aforementioned experiments, I suggest the authors use a viral antigen-specific antibody to label virus-infected cells, rather than an anti-dsRNA. IF staining using a viral antigen-specific antibody is a more convincing data and usually the image quality will be greatly improved. In addition, the authors may show the distribution/abundance of MYADM in wildtype and knockout organoids. In Fig. 2a, not all dsRNA signals are within the cytosol, which is not reasonable.

We appreciate the reviewer's suggestion to use a viral antigen-specific antibody for labeling virus-infected cells instead of anti-dsRNA. To address concerns about dsRNA signal localization, we have increased the brightness of the F-actin channel in the images (illustrated below) to clarify that all dsRNA signals are indeed confined to the cytosol. It is important to note that PeV-A1 infection could result in cell extrusion from the organoids following severe replication, leading to weak signals at cell edges. The DAPI staining shows the nuclei of the extruded cells.

In response to the reviewer's recommendation, we have tested a recently available PeV specific antibody (anti-VP1) for immunostaining. The results demonstrated a more expansive signal in the cytosol compared to the anti-dsRNA antibody reflecting the broader distribution of the capsid

protein in the cytosol as compared to viral dsRNA, which is restricted to replication organelles (illustrated below). Reassuringly, the number of infected cells in wild-type cells and the absence of staining in infected MYADM knockout cells was similar with both antibodies. Also, the anti-VP1 showed the cell extrusion pattern following severe infection. We have opted to maintain the anti-dsRNA staining in the manuscript, as it more directly stains active viral RNA replication.

Wild type (WT) human colonoid infected with PeV-A1 and stained with anti-dsRNA or anti-VP1 antibody.

To show the abundance of MYADM in wildtype and knockout organoids, we have performed western blot analysis confirming the knockout of MYADM in MYADM KO organoids (Extended Data Figure 6). Additionally, we attempted immunostaining using the same MYADM antibody for western blot. However, the antibody resulted in nonspecific staining in both cell lines and organoids. Subsequently, we tested three commercially available antibodies for MYADM (Abcam ab212997; Bioss, BS-19143R; and Mybiosource, MBS669098) and none of them proved effective for immunostaining.

Extended Data Figure 7. Validation of human colonoids MYADM isogenic knockout. b, Western blot analysis of MYADM expression in organoids.

MYADM binding to PeV-A1 particles was dependent on slightly acidic conditions representative of early and late endosomes as shown by cell-free immunoprecipitation assay. It would be more convincing if the authors could show data using endosomal acidification inhibitors (such as chloroquine, bafilomycin A1 and NH₄CL) in cell lines, and ideally organoids.

Indeed, the pH dependence for MYADM binding is mainly based on the *in vitro* experiments. We have also observed binding in a cell entry assay IP experiment at early time-points. Although it is likely that the time points correspond to uptake into the endocytic system, which is known to lead to acidification, we agree with the reviewer that additional experiments would be helpful. A similar point was made by reviewer 2. Following the reviewers' recommendation, we conducted time course immunoprecipitation assays (IP) with endosomal acidification inhibitors, bafilomycin A1

(BafA1) and NH₄CL, in Vero cells. The same IP assay was performed with samples after 15 min internalization, but pretreated with 100 nM BafA1 or 30 mM NH₄CL for 1 hour at 37 °C and along the whole process. While the results confirmed the binding of PeV-A1 protein with MYADM during entry, we observed no significant difference in inhibitor-treated cells (shown below). Although this could mean that endocytic acidification in cells is not required for MYADM's interaction with PEV-A1, the observation that *in vitro* the interaction already occurs at pH-6.5 and is optimal at pH-6.0 leaves open the possibility that BafA1 or NH₄CL treatment did not prevent the slight acidification typical for early endosomes. For example, in a previous study with A431 cells, the intralysosomal pH increased from about 5.1-5.5 to about 6.3 in the presence of BafA1 (PMID: 1832676). The lack of tools to completely prevent acidification (and the likely cell toxicity that would be associated with that) limits our ability to perform these experiments. To avoid the impression that we have conclusively proven the pH dependence in cells, we have removed the indication of the endosomal pH in the schematic model in fig. 4 and we have clarified in the discussion what we can conclude from the current data: "We demonstrated that MYADM binds to PeV-A1 particles and is essential for functional viral entry, which points to a role as receptor. MYADM knockout did not reduce cell binding, suggesting that it does not act as a primary cellular attachment factor for PeV-A1. Rather, binding of PeV-A1 to MYADM during viral entry was enhanced on raising the temperature to allow endocytosis. Moreover, the *in vitro* binding between MYADM with parechovirus particles was detected at slightly acidic conditions (pH ≤ 6.5). Whether during a viral infection the binding is similarly dependent on the pH will need to be experimentally validated because *in vivo* and *in vitro* binding conditions can differ."

Time point virus particle binding assay. Vero cells and Vero cells expressing HA-tagged human MYADM (Vero + WT) were pretreated with 100 nM BafA1 or 30 mM NH₄CL for 1 hour at 37 °C, incubated with concentrated PeV-A1 at 4 °C for 1 h, followed by incubation at 37 °C for 15 min after removing unbound viruses along with the compounds. Cells were lysed (Input) and immunoprecipitated with anti-HA beads (IP). Viral RNA levels were measured by RT-qPCR.

Minor issues

1. Line 101. The authors used apical-out intestinal organoids for infection experiments and cited the paper. However, they may not describe the preparation of these apical-out organoids in Methodology.

The descriptions for preparing apical-out organoids were in the section titled “Virus infection in human colonoids”. We have now addressed this specifically by adding the subtitle “Virus infection of apical-out human colonoids.”

2. Line 99. “cell models” is not a good description for organoids. consider using “biologically relevant in vitro models”.

We have replaced “cell models” in line 99 with “biologically relevant in vitro models”, as suggested by the reviewer.

REVIEWERS' COMMENTS

Reviewer #1 (Remarks to the Author):

The authors have adequately addressed all of my comments and concerns.

Reviewer #3 (Remarks to the Author):

The authors have adequately addressed my inquiries.

Reviewer #4 (Remarks to the Author):

The authors describe the identification of MYADM as a receptor for entry for parechoviruses. The authors have discussed and rebuttled the reviewers comments well. However in the rebuttal and manuscript it seems that MYADM is not a receptor for parechoviurses as the authors suggest in the title but rather a host factor involved in virus entry. Therefore the title should be changed to better reflect the data